# Improved methods for marking active neuron populations

Benjamien Moeyaert [1], Graham Holt[1,8], Rajtarun Madangopal [2], Alberto Perez-Alvarez [3], Brenna C. Fearey [3], Nicholas F. Trojanowski [4], Julia Ledderose[5], Timothy A. Zolnik[6], Aniruddha Das[7], Davina Patel[7], Timothy A. Brown[1], Robert N.S. Sachdev[6], Britta J. Eickholt[5,6], Matthew E. Larkum[6], Gina G. Turrigiano[4], Hod Dana [1,7], Christine E. Gee [3], Thomas G. Oertner [3], Bruce T. Hope[2] & Eric R. Schreiter[1]

Marking functionally distinct neuronal ensembles with high spatiotemporal resolution is a key challenge in systems neuroscience. We recently introduced CaMPARI, an engineered fluorescent protein whose green-to-red photoconversion depends on simultaneous light exposure and elevated calcium, which enabled marking active neuronal populations with single-cell and subsecond resolution. However, CaMPARI (CaMPARI1) has several drawbacks, including background photoconversion in low calcium, slow kinetics and reduced fluorescence after chemical fixation. In this work, we develop CaMPARI2, an improved sensor with brighter green and red fluorescence, faster calcium unbinding kinetics and decreased photoconversion in low calcium conditions. We demonstrate the improved performance of CaMPARI2 in mammalian neurons and in vivo in larval zebrafish brain and mouse visual cortex. Additionally, we herein develop an immunohistochemical detection method for specific labeling of the photoconverted red form of CaMPARI. The anti-CaMPARI-red antibody provides strong labeling that is selective for photoconverted CaMPARI in activated neurons in rodent brain tissue.

[1] Howard Hughes Medical Institute, Janelia Research Campus, 19700 Helix Drive, Ashburn, VA 20147, USA. [2] Neuronal Ensembles in Addiction Section, Behavioral Neuroscience Research Branch, Intramural Research Program, National Institute on Drug Abuse, National Institutes of Health, Baltimore, MD 21224, USA. [3] Institute for Synaptic Physiology, University Medical Center Hamburg-Eppendorf, 20251 Hamburg, Germany. [4] Department of Biology, Brandeis University, Waltham, MA 02454, USA. [5] Institute of Biochemistry, Charité - Universitätsmedizin Berlin, 10117 Berlin, Germany. [6] NeuroCure Cluster of Excellence, Department of Biology, Humboldt University, 10117 Berlin, Germany. [7] Department of Neurosciences, Lerner Research Institute, Cleveland Clinic Foundation, Cleveland, OH 44195, USA. [8] Present address: Program in Computational Biology and Bioinformatics, Duke University, Durham, NC 27708, USA. Correspondence and requests for materials should be addressed to E.R.S. (email: schreitere@janelia.hhmi.org)

The coordinated activity of neurons that are spatially distributed throughout complex tissues like the brain are thought to mediate critical functions such as the selection and generation of actions in response to stimuli, learning from the outcomes of those actions, and the storage and recall of memories of those actions and outcomes. Methods to identify these neuronal ensembles based on their activity over various time and spatial scales are critical to furthering our understanding of brain function.

Activity-dependent genes such as the immediate early genes (IEGs) Fos and Arc[1] have been extensively used to mark and manipulate recently activated neuronal ensembles[2–5]. However, IEG-based methods suffer from poor temporal resolution (minutes to hours)[6,7] and the relationship between neuronal activity and IEG expression is indirect. Some of these issues can be circumvented by imaging neuronal calcium transients in vivo in behaving animals, for instance, using head-fixed two-photon microscopy through cranial windows[8] or epifluorescent microscopy through microendoscopic lenses[9,10]. Calcium has a direct and quantifiable relationship with electrical spiking in neurons[11]. However, live calcium signals can only be imaged with limited fields of view, and it is challenging to correlate these signals with post hoc in vitro analyses, such as immunohistochemistry or gene expression profiling.

As a new approach to examining active neuronal ensembles, we recently introduced CaMPARI, a photoconvertible green fluorescent protein whose irreversible photoconversion (PC) to a red form is not only dependent on the presence of light but also on the concentration of free calcium ions[12]. However, this early version of CaMPARI (CaMPARI1) had some shortcomings, including a modest PC contrast, slow calcium unbinding, and sensitivity to chemical fixatives such as formaldehyde[13].

In this work, we present CaMPARI2, an improved variant of CaMPARI1. Using site-directed amino acid mutagenesis combined with functional screening and selection, we significantly increase the contrast of green-to-red PC between the calcium-bound and calcium-free states. This effect is further enhanced by a higher brightness of the red form of the protein. CaMPARI2 also has a higher rate of calcium unbinding compared to CaMPARI1. To accommodate different cell types and calcium levels, we develop a range of affinity variants, with dissociation constants ($K_d$) for calcium binding ranging from 100 nM to 1 μM. We demonstrate the functionality of CaMPARI2 in vitro and in vivo. Additionally, we develop a monoclonal antibody that specifically recognizes the photoconverted red form, allowing immunohistochemical detection and selective amplification of the red CaMPARI2 signal in fixed cells or tissue.

## Results

**Engineering CaMPARI2.** We sought to improve the brightness of the green and red forms of CaMPARI1 and improve the difference in PC rate between high and low calcium conditions to enhance the visible contrast between active and non-active cells in a neuronal population. We targeted site-saturation mutagenesis to amino acid positions around the fluorescent protein chromophore and at the protein interface between the fluorescent protein and the calcium-sensitive domains of CaMPARI, as defined by the crystal structures of CaMPARI and other genetically encoded calcium inhibitors[12,14,15]. We assayed fluorescence and PC contrast (extent of PC in high-calcium vs. low-calcium conditions) of ~950 unique single amino acid substitutions at 50 separate positions of CaMPARI1_W391F-V398L in a medium-throughput assay in bacterial lysates[12]. CaMPARI1_W391F-V398L was chosen as a template for optimization due to its four times faster calcium release kinetics and reasonable $K_d$ for $Ca^{2+}$ (~ 200 nM).

Several single amino acid variants improved CaMPARI brightness, PC contrast, or both (Supplementary Table 1). A library of combinations of these single amino acid variants was then screened in the same way, which revealed variants with further improved brightness and PC contrast (Supplementary Table 2).

One variant with five amino acid substitutions was selected that showed significantly higher fluorescence in both the green and red form, nearly four-fold higher PC contrast, and exhibited a $K_d$ for calcium of 285 nM (CaMPARI1_W391F-V398L-Q142V-F198Y-C202T-L217I-N345S (Fig. 1a, Supplementary Fig. 1, Supplementary Table 2)). To enable a wide range of applications in different cell types and organisms with various baseline calcium levels, we mutated key residues in the calmodulin-binding peptide at positions that were previously identified to alter the calcium affinity of CaMPARI[12]. We generated 15 variants with $K_d$ values between 145 nM and 2.1 μM (Supplementary Table 3) and selected four that spanned this affinity range (Supplementary Table 4) to characterize further. Three consecutive epitope tags (FLAG–HA–myc) were added to the C-terminus of the optimized CaMPARI and its affinity variants, enabling flexible immunohistochemical detection in tissue. CaMPARI1_W391F-V398L-Q142V-F198Y-C202T-L217I-N345S_FLAG-HA-myc is hereafter referred to as CaMPARI2 (Supplementary Fig. 1). Addition of the C-terminal epitope tags led to increased calcium affinity (Supplementary Fig. 2), although the reasons for this observation are unclear.

**Characterization of CaMPARI2.** We characterized purified CaMPARI2 protein in vitro (Table 1, Fig. 1, Supplementary Fig. 3 and 4) and found that the green and red forms were 60% and 130% brighter, respectively, compared with CaMPARI1 and that calcium unbinding occurred several times faster. Although the rate of PC in high calcium conditions is similar for CaMPARI1 and CaMPARI2 ($0.020\,s^{-1}$ vs. $0.026\,s^{-1}$), CaMPARI2 has a markedly lower PC rate in its calcium-free state ($0.00022\,s^{-1}$) compared to CaMPARI1 ($0.0010\,s^{-1}$) (Fig. 1 and Supplementary Table 5). Thus CaMPARI2 exhibits a six-fold larger PC rate contrast (PC rate with calcium/PC rate without calcium) compared to CaMPARI1. We screened a CaMPARI2 library with site-saturated mutations at ~50 positions for higher PC rate when saturated with calcium but did not identify significantly improved variants.

Next, CaMPARI1 and CaMPARI2 were compared in dissociated primary rat hippocampal neurons in culture. To compare PC rates, we shined 405 nm light through a widefield microscope objective while driving action potential firing at 80 Hz and followed the PC over time (Fig. 1c, right). As we saw with purified proteins, the rates of PC of CaMPARI1 and CaMPARI2 during action potential firing (high calcium) were similar ($0.22\,s^{-1}$ vs. $0.14\,s^{-1}$, respectively), but the rate of PC in resting neurons (low calcium) was five times lower for CaMPARI2, resulting in a three-fold higher PC rate contrast (Supplementary Fig. 5 and Supplementary Table 5). We next measured the extent of PC of CaMPARI1, CaMPARI2, and four affinity variants following different frequencies of action potential firing during a 2 s light pulse. As expected, the highest affinity variant (CaMPARI2_F391W) reached maximum PC at lower stimulation frequencies, while the lowest affinity variant (CaMPARI2_L398T) required higher stimulation frequencies (Fig. 1d). CaMPARI1 exhibited relatively high amounts of PC in resting neurons, resulting in a low PC contrast at all stimulation frequencies. Finally, we saw that CaMPARI2 has faster calcium unbinding kinetics in neuron cultures (Supplementary Fig. 6). The relative rates of calcium unbinding between the different affinity mutants depended on the number of evoked action

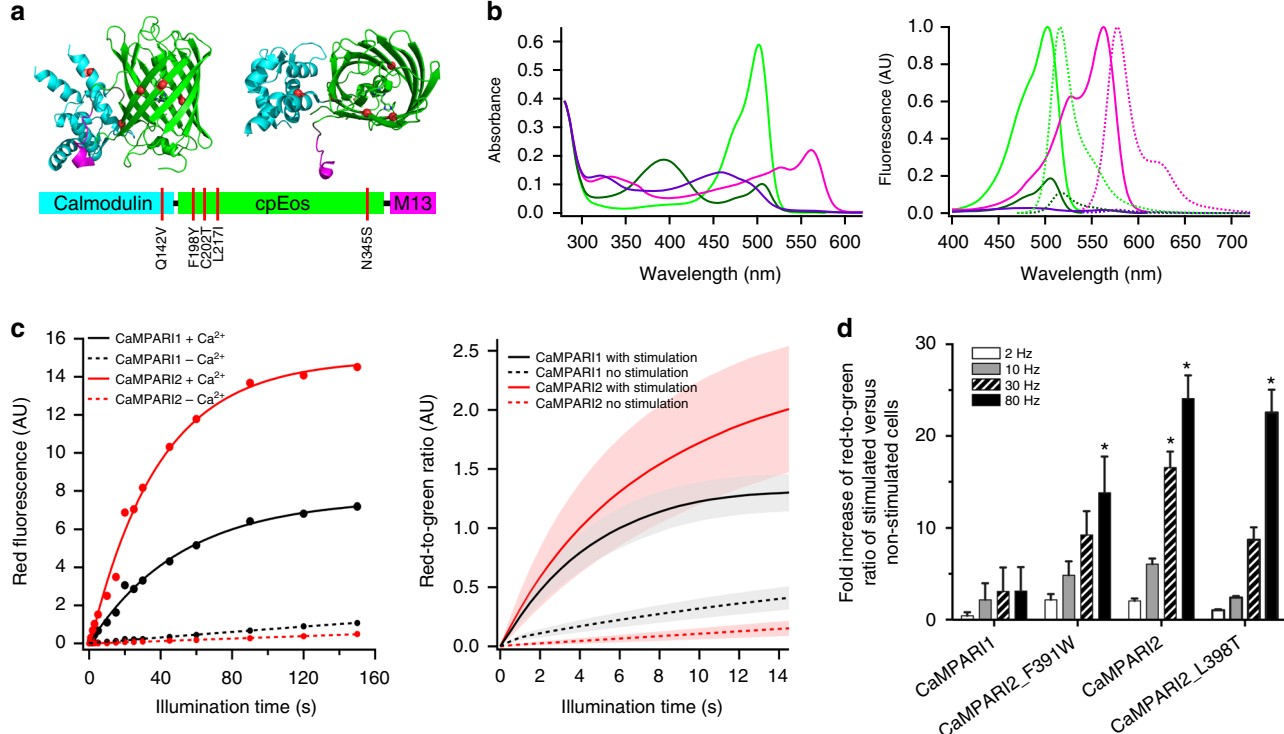

**Fig. 1** In vitro characterization of CaMPARI2. **a** Primary (bottom) and tertiary (top) structures of CaMPARI2. Mutations relative to CaMPARI1_W391F-V398L are shown in red. Two orthogonal views of the same CaMPARI crystal structure (PDB ID 4OY4 [https://doi.org/10.2210/pdb4OY4/pdb] are shown. **b** Absorption (left) and fluorescence (right) excitation (full line) and emission (dotted line) spectra of CaMPARI2. Green and magenta spectra represent the green and red forms of CaMPARI; bright and dark lines represent the calcium-free and calcium-saturated states. **c** Photoconversion timecourse showing the red fluorescence of CaMPARI1 (black) and CaMPARI2 (red) as a function of exposure to 405 nm light. Left: photoconversion of purified CaMPARI protein in the presence (solid lines) or absence (dashed lines) of calcium. Right: photoconversion of primary rat hippocampal neurons with (solid lines) or without (dashed lines) 80 Hz stimulation. **d** Fold increase of the red-to-green ratio of CaMPARI variant-expressing neurons after 2 s of photoconversion during different electrical stimulation frequencies relative to no stimulation. Error bars are standard deviation, $n = 3$, asterisks denote values significantly differing from the corresponding CaMPARI1 value ($p < 0.001$, Dunnett's multiple comparisons test)

**Table 1 Photophysical properties of CaMPARI1, CaMPARI2, CaMPARI2_F391W, and CaMPARI2_L398T**

| | $\lambda_{ex}$, G (nm) | $\lambda_{em}$, G (nm) | $\lambda_{ex}$, R (nm) | $\lambda_{em}$, R (nm) | $\varepsilon$, G (mM$^{-1}$ cm$^{-1}$) | $\varepsilon$, R (mM$^{-1}$ cm$^{-1}$) | QY, G (%) | QY, R (%) | Brightness[a], G | Brightness[a], R | $K_d$ Ca$^{2+}$ (nM) | $\Delta F/F$ | $k_{off}$ (s$^{-1}$) |
|---|---|---|---|---|---|---|---|---|---|---|---|---|---|
| CaMPARI1 | 498 | 514 | 560 | 576 | 73.5 ± 5.8 | 32 | 78 | 58 | 1.0 | 1.0 | 134.7 ± 10.8 | 5.9 ± 0.2 | 0.29 ± 0.017 |
| CaMPARI2 | 502 | 516 | 562 | 577 | 111.3 ± 2.2 | 65 | 81 | 65 | 1.6 | 2.3 | 199.2 ± 11.8 | 7.8 ± 0.2 | 1.43 ± 0.020 |
| CaMPARI2_F391W | 502 | 516 | 562 | 577 | 114.8 ± 1.7 | 60 | 81 | 62 | 1.6 | 2.0 | 109.7 ± 2.7 | 6.8 ± 0.3 | 0.59 ± 0.082 |
| CaMPARI2_L398T | 502 | 516 | 562 | 577 | 114.2 ± 3.6 | 58 | 80 | 66 | 1.6 | 2.1 | 824.6 ± 26.2 | 5.4 ± 0.8 | 2.50 ± 0.063 |

$\varepsilon$ is the extinction coefficient in mM$^{-1}$ cm$^{-1}$
[a]Brightness is expressed as molecular brightness (extinction coefficient × quantum yield) normalized to CaMPARI1 in the corresponding state. Full table can be found as Supplementary Table 4. ± values are SD from $n = 3$ or 4 ($n = 15$ for $k_{off}$)

potentials and were different from the stopped flow experiments with purified proteins (Table 1, Supplementary Table 4). Nevertheless, CaMPARI2 and its variants generally exhibited faster calcium unbinding kinetics than CaMPARI1, and tighter calcium binding correlated with slower calcium unbinding. We also confirmed that the extent of PC depended on the timing between the stimulus and the light pulse in a way that reflected calcium unbinding kinetics (Supplementary Fig. 7).

We next compared CaMPARI1 to CaMPARI2 (CaMPAR-I2_F391W-L398V without epitope tags, see Supplementary Table 4) in CA1 neurons of organotypic rat hippocampal slice cultures during whole-cell patch clamp electrophysiology (Fig. 2).

Similar to our findings in dissociated neuron cultures, we found that CaMPARI2 is brighter and has a 2.5-fold higher PC rate contrast compared to CaMPARI1 (Supplementary Table 5). After photoconverting during synaptic stimulation, the red signal is strongly visible for at least 24 h, but the R/G ratio falls to ~20% of its maximum value after 72 h, allowing re-labeling by repeating the stimulation and PC (Supplementary Fig. 8).

The PC signal was then compared in vivo in larval transgenic zebrafish expressing either CaMPARI1 or CaMPARI2 from a neuron-specific promoter. When neuronal activity was blocked with the sodium channel blocker tricaine during a PC light pulse, the red-to-green (R/G) ratios were slightly lower with

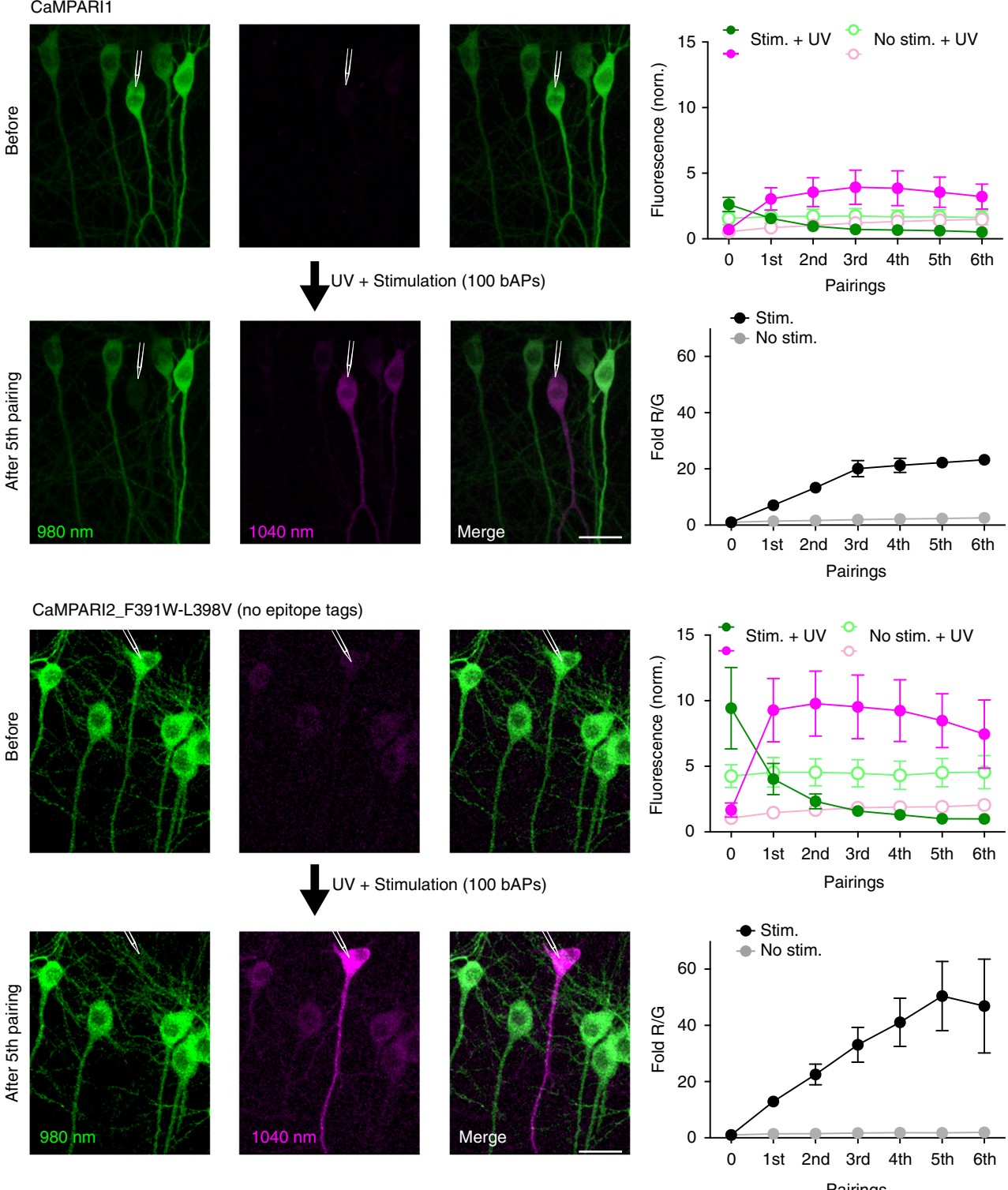

**Fig. 2** Brightness and photoconversion of CaMPARI1 and CaMPARI2. Two-photon images of CaMPARI1 and CaMPARI2_F391W-L398V (no epitope tags) expressed in CA1 neurons of rat hippocampal slice cultures before and after pairing of stimulation (100 bAPs at 100 Hz) with UV light (2 s, 16 mW mm$^{-2}$). Two lasers were simultaneously employed to acquire images of the green (980 nm) and red species (1040 nm) of the CaMPARI variants. A single cell, denoted by a pipette drawing, is patched and stimulated during the UV illumination. Green and red fluorescence is quantified along with the fold R/G in both stimulated (CaMPARI1 $n = 5$; CaMPARI2 $n = 6$) and unstimulated neighboring neurons (CaMPARI1 $n = 5$; CaMPARI2 $n = 14$) as a function of the number of pairings. Error bars are SEM. Note that fluorescence intensity is normalized to the laser power (see Methods), showing the increased brightness of CaMPARI2. Scale bars are 25 μm

CaMPARI2, in agreement with a lower baseline PC rate for CaMPARI2 (Fig. 3, Supplementary Fig. 9, 10). After PC of freely swimming fish to mark neurons with ongoing spontaneous activity, we measured many neurons with higher red/green ratios and a larger range of red/green ratios with CaMPARI2 (Supplementary Fig. 9 and 10), suggesting that it is possible to mark activity on a fine scale in the CaMPARI2 transgenic zebrafish.

Finally, to demonstrate the ability of CaMPARI2 to mark neurons in response to specific stimuli in vivo, we expressed CaMPARI2 and CaMPARI2-F391W in the mouse visual cortex and measured CaMPARI2 PC as well as traditional calcium indicator fluorescence in response to specific visual stimuli. Bright CaMPARI2 labeling was evident 15 days after adeno-associated virus (AAV) injection and individual layer 2/3 neurons could be easily identified (Fig. 4a). Lightly anesthetized mice were presented with upward drifting gratings while illuminated with PC light. Following the PC, the calcium response of individual neurons was recorded in response to the presentation of drifting gratings in eight different directions (Fig. 4b, Methods).

We identified neurons with significant calcium responses (analysis of variance (ANOVA) test, $p < 0.01$) during presentation of any of the visual stimuli and grouped them as neurons with a significant calcium response during the upward drifting gratings displayed during PC (PC-tuned) and with no significant response to the upward drifting grating but with significant responses to other directions of motion (responsive but not PC-tuned). In addition, we calculated the orientation selectivity index (OSI)[16] of each neuron. We grouped together all responsive neurons with OSI < 0.5 (broadly tuned, Fig. 4c). This group of cells with broad-tuning to grating orientation contains most of the inhibitory cells[16–18]. It was previously shown that inhibitory neurons have

lower response amplitudes during calcium imaging[17] and lower OSI for responding to drifting grating stimuli than excitatory neurons[16,18], allowing us to categorize them separately based on their response profile.

Of all the segmented neurons, 11.5% were identified as responsive (48/359 and 18/203 and 41/525, 48/366, and 63/419 for two CaMPARI2 and three CaMPARI2-F391W mice, respectively) and 3.2% was PC-tuned (10 and 1 and 17, 26, and 6 for two CaMPARI2 and three CaMPARI2-F391W mice, respectively). The R/G ratio of the responsive cells was correlated to the peak change in fluorescence for the northward moving grating stimulus (Fig. 4c, Supplementary Fig. 11 top) but not to the orthogonal direction stimulus (Supplementary Fig. 11 middle). The R/G ratio was significantly higher for PC-tuned cells than both responsive and not PC-tuned cells and cells that had no significant change to the visual stimuli (Fig. 4d, Supplementary Fig. 11 bottom, Wilcoxon Rank-Sum Test), demonstrating that CaMPARI2 can mark neurons that are responsive to specific stimuli in vivo. We note that the PC of CaMPARI2 yielded better separation among these groups than CaMPARI2-F391W (Supplementary Fig. 11), presumably because the higher calcium affinity of CaMPARI2-F391W led to partial saturation in the absence of stimuli that resulted in higher baseline PC and lower contrast between responsive and non-responsive neurons.

**Development of an anti-CaMPARI-red antibody.** Although the red form of CaMPARI2 is bright and easily detectable in live cells and tissue with conventional fluorescence microscopy, chemical tissue fixation using formaldehyde, for example, generally results in loss of fluorescent signal due to changes in protein conformation following chemical modification by formaldehyde. To

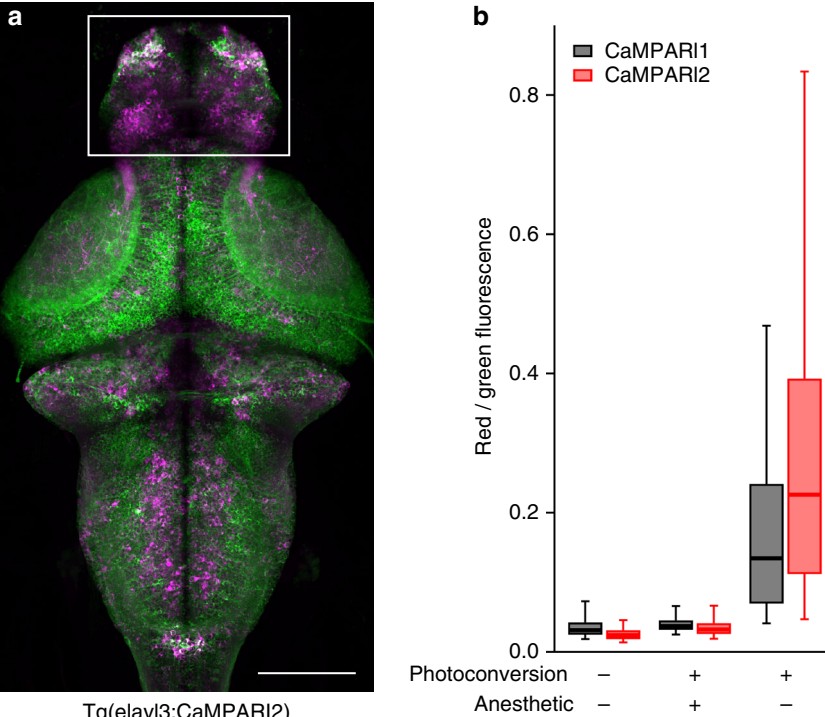

**Fig. 3** In vivo characterization of CaMPARI2 in zebrafish. **a** Representative *z*-projection from a confocal stack from a 6-dpf larval transgenic pan-neuronal CaMPARI2 zebrafish brain photoconverted for 30 s during free swimming. **b** Boxplots represent the distribution of red-to-green fluorescence signals from individual neurons (between 1800 and 6000 cells per condition, representing 2–4 fish (Supplementary Fig. 9, 10). Data are measured from neurons in the forebrain (white box in **a**) following photoconversion of either freely swimming or tricaine-anesthetized larval zebrafish. Box represents 1st, 2nd, and 3rd quartile, while whiskers represent the 5th and 95th percentile. Scale bar is 100 μm

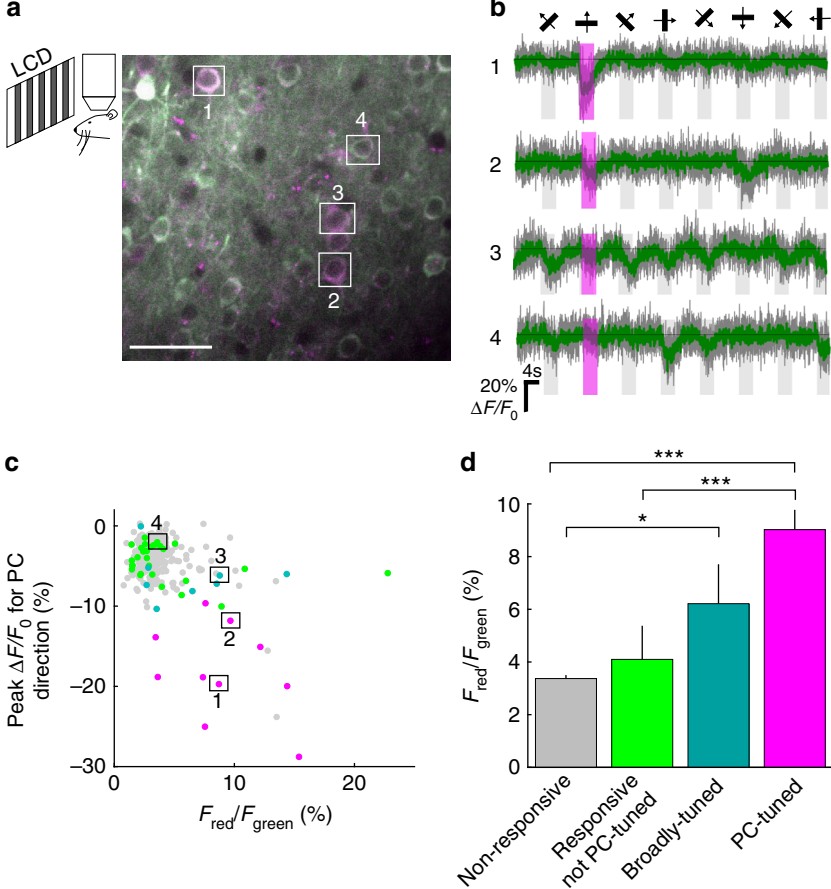

**Fig. 4** CaMPARI2 activity and PC in mouse primary visual cortex. **a** Schematic of the experimental setup (left). Two-photon fluorescence signal from cortical layer 2/3 neurons in V1 after PC (right). **b** Changes in fluorescence signal from four example cells marked in **a**. Cell numbers 1–2 are PC-tuned and have high OSI (0.8 and 0.92, respectively), cell number 3 is responsive and broadly tuned (OSI = 0.22), and cell number 4 is responsive and not PC-tuned (OSI = 0.96). **c** Correlation between peak $\Delta F/F_0$ for the northward moving grating stimulus and the red-to-green ratio for individual responsive cells expressing CaMPARI2. Cells were grouped into four categories: non-responsive (not significantly responsive cells, gray dots), broadly tuned (significantly responsive and OSI < 0.5, cyan), PC-tuned (OSI > 0.5 and significant response to northward moving grating stimulus, magenta), and not PC-tuned (OSI > 0.5 and no significant response to northward moving grating stimulus, green). **d** Comparison of red-to-green ratio distribution of the four groups mentioned in **c**. PC efficiency was higher for PC-tuned cells, leading to a significant increase in red-to-green ratio. *$p < 0.05$, ***$p < 0.001$ (Wilcoxon Rank-Sum Test). Error bars indicate the standard error. Scale bar is 40 μm

improve the signal-to-noise ratio when imaging CaMPARI in fixed tissue, we sought to develop a monoclonal antibody that specifically targets the photoconverted form of CaMPARI and allows recovery or amplification of the red signal.

To do so, we purified and photoconverted EosFP protein, subjected it to proteolysis, and further purified the proteolytic fragment containing the red chromophore (Fig. 5a and Supplementary Fig. 12). This proteolytic fragment, which is identical to the equivalent region of CaMPARI1 and CaMPARI2, was used as an antigen to generate a mouse monoclonal antibody ("anti-CaMPARI-red") using standard protocols. Western blots containing several fluorescent proteins demonstrate the specificity of the anti-CaMPARI-red antibody for the photoconverted red forms of green-to-red photoconvertible fluorescent proteins, but not the green form (Supplementary Fig. 13). Other commonly used red fluorescent proteins (such as mCherry, mNeptune, mRuby) with different chromophore structures were not recognized by anti-CaMPARI-red. The antibody amino acid sequence was determined and is given in Supplementary Fig. 14.

Following PC in vivo, we performed immunohistochemistry on mouse and rat brain slices expressing CaMPARI2 to confirm the

functionality and specificity of anti-CaMPARI-red antibody labeling in complex tissue. Similar to cell culture experiments, the endogenous green and especially red CaMPARI fluorescence decreased significantly after fixation but could be recovered with antibody staining (Figs. 5b, 6 and Supplementary Fig. 15, 16 and 17).

In both cultured cells and rodent brain tissue, the anti-CaMPARI-red antibody specifically recognizes the photoconverted form of CaMPARI and the stained images are reflective of the endogenous red CaMPARI2 signal in vivo (Figs. 5b, 6 and Supplementary Fig. 16, 17), although staining of dissected larval zebrafish brains with anti-CaMPARI-red antibody was unsuccessful in our hands. The immunohistochemical labeling correlates with the endogenous red fluorescent signal of the photoconverted CaMPARI2 molecules (Fig. 6) and the ratio of the intensity of the anti-CaMPARI-red signal to the anti-FLAG epitope tag signal (total CaMPARI) correlates with the endogenous R/G fluorescence ratio of CaMPARI. In other words, the immunohistochemical labeling reflects the relative extent of PC and the endogenous R/G fluorescence intensity ratio of CaMPARI in multiple cell types (Fig. 6).

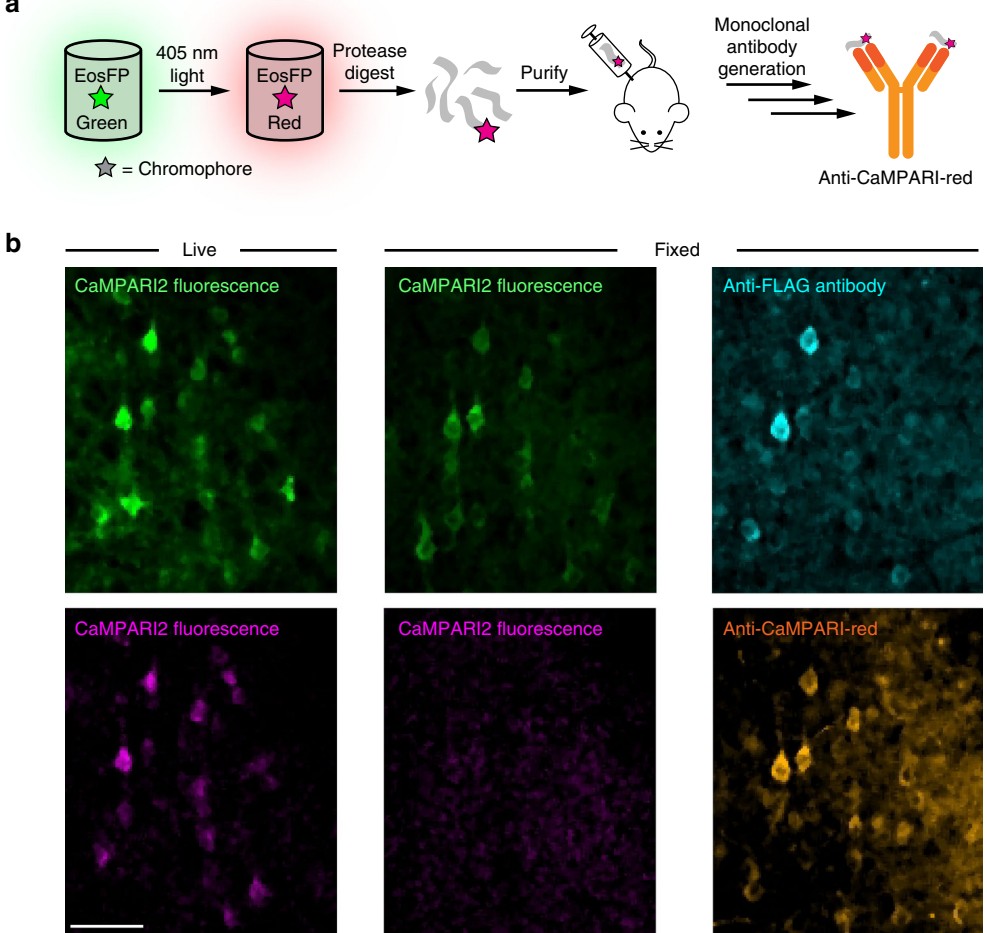

**Fig. 5** Anti-CaMPARI-red antibody. **a** Schematic representation of the protocol used to generate the anti-CaMPARI-red monoclonal antibody. **b** Green (top) and red (bottom) endogenous CaMPARI2 fluorescence (left and middle columns) in mouse brain tissue imaged before (left column) and after (middle column) chemical fixation with paraformaldehyde. The right column shows the same region of fixed tissue after antibody staining with anti-FLAG antibody for total CaMPARI (top, cyan) and anti-CaMPARI-red antibody (bottom, orange). Note that following chemical fixation much of the red CaMPARI fluorescence was lost but was recovered with the anti-CaMPARI-red antibody staining. Scale bar is 50 μm

## Discussion

As a complement to existing techniques for marking active neuronal populations, we recently introduced CaMPARI, a fluorescent protein whose green-to-red PC is calcium-dependent. CaMPARI allows marking of active neurons with finer time resolution than activity-dependent gene expression and provides a more permanent signal than transient calcium indicators like GCaMP. In this work, we have developed CaMPARI2, with higher molecular brightness, faster calcium unbinding kinetics, and less background PC in the absence of calcium compared to CaMPARI1, leading to a higher contrast between high-calcium and low-calcium cells. CaMPARI2 is therefore an improved tool for a range of applications. For instance, the reduced background PC is useful for integrating sparse neuronal activity, which calls for long illumination times. Second, the 130% higher brightness of the red form of CaMPARI2 compared to the red form of CaMPARI1 facilitates imaging and image processing. This should allow for detection of smaller amounts of photoconverted CaMPARI from smaller structures within complex tissue. The slow CaMPARI turnover dramatically increases the potential field of view for live imaging experiments. Distant neurons that are active at an instant of time can be marked by PC and then mapped at high resolution post hoc without time constraints. Relabeling will allow chronic imaging of active neuronal ensembles.

Additionally, the faster calcium unbinding kinetics increases the temporal resolution of calcium activity integration, allowing for more precise PC of neurons active during a short epoch of animal behavior. A corollary to faster calcium release kinetics of CaMPARI2 is that, during each calcium transient, less time is spent in the state that photoconverts at a high rate. This leads to less absolute PC of CaMPARI2 relative to CaMPARI1 for an equivalent number of calcium transients and equivalent light exposure. This is largely offset by the brighter red fluorescence per molecule of CaMPARI2 but could be limiting under some circumstances. Our experiments in larval zebrafish brain showed a similar amount of red fluorescence with CaMPARI1 and CaMPARI2 and a wide distribution of red/green values among CaMPARI2-expressing neurons, indicating that it can delineate fine gradations of activity. In vivo calcium imaging and PC in mouse visual cortex confirms that CaMPARI2 is able to selectively label neurons that are active during a specific stimulus. We identified variants of CaMPARI2 with higher or lower calcium affinity and variable calcium release kinetics (Supplementary Table 4) that might be preferable in other tissues or cell types.

Many conditions for chemical fixation and preservation of tissue lead to unfolding of proteins, which results in a loss of fluorescent signal from fluorescent proteins. To circumvent this problem and to amplify the signal from the red photoconverted

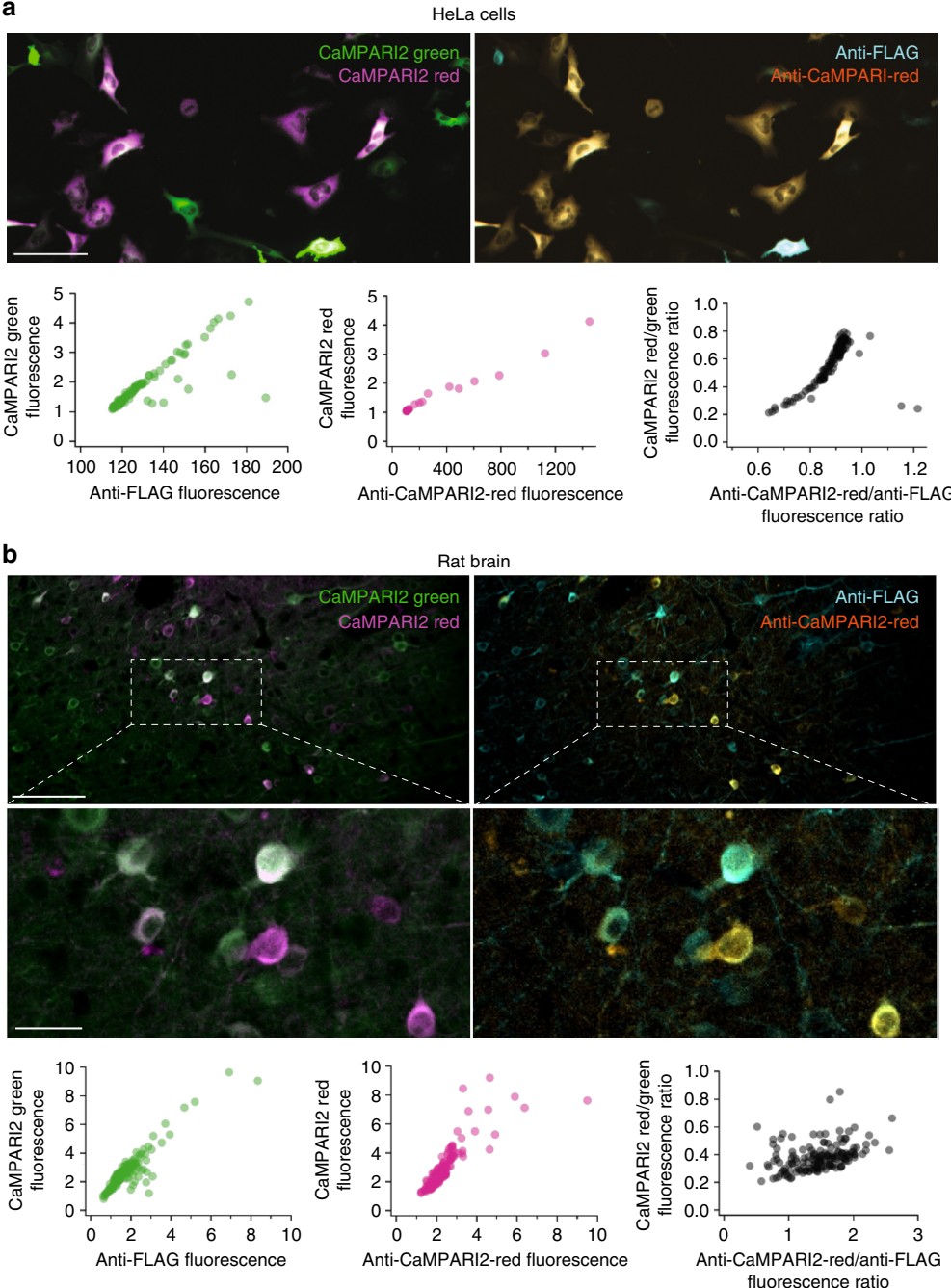

**Fig. 6** Comparison between endogenous fluorescence and anti-CaMPARI-red antibody stain. HeLa cells (**a**) and rat brain (**b**) were immunostained with anti-CaMPARI2-red and anti-FLAG antibody as described in Supplementary Methods. Plots of endogenous fluorescence vs. the antibody signal show a linear relationship (green and magenta scatterplots). The black scatterplots show the correlation between the endogenous red-to-green ratio and the anti-CaMPARI-red-to-anti-FLAG ratio. Scale bars are 100 or 25 μm (insets)

form of CaMPARI, we developed a mouse monoclonal antibody that specifically detects the red, but not green, CaMPARI chromophore. It also recognizes the red chromophores of other common green-to-red photoconvertible fluorescent proteins (such as EosFP, Kaede, and mMaple) but does not recognize conventional red fluorescent proteins (such as mCherry), which have a different chromophore structure. Although the chromophore of CaMPARI and other fluorescent proteins is normally buried within the folded structure of the protein and is thus not accessible for antibody recognition, we hypothesize that protein unfolding during chemical fixation exposes the chromophore for binding by the antibody.

In summary, our current work expands and improves the toolkit for marking active neuronal populations in behaving animals. CaMPARI2 offers improved brightness and contrast over CaMPARI1, and we recommend its adoption for most preparations. Additionally, the anti-CaMPARI-red antibody offers additional flexibility by enabling immunohistochemical detection of CaMPARI PC, bringing back the signal when endogenous fluorescence is lost during chemical fixation of tissue.

## Methods

**Directed evolution of CaMPARI2 and in vitro characterization**. We conducted multiple rounds of site-saturation mutagenesis, functional screening, and selection to improve the properties of CaMPARI. Site-saturation mutagenesis at individual amino acid positions was done using the QuikChange Multi protocol (Agilent). Generally, 90 colonies were picked from a site-saturation mutagenesis library at each amino acid position, along with controls, into deep-well 96-well blocks. The T7 Express *Escherichia coli* bacteria (New England Biolabs) were grown at 30 °C for 36 h and pelleted by centrifugation. Soluble lysate was prepared from the pellets by incubation with Bacterial Protein Extraction Reagent (Thermo Fisher) followed by centrifugation. Functional screening included measurement of green and red fluorescence of bacterial lysates using a fluorescence plate reader (Tecan) after addition of 0.5 mM $CaCl_2$ or 1 mM EGTA to separate lysate aliquots. Fluorescence was measured again after irradiation with 405 nm light using an light-emitting diode (LED) array (Loctite; 1 min, ~200 mW $cm^{-2}$) and again after addition of 10 mM EGTA and 5 mM $CaCl_2$, respectively. From these fluorescence reads, we selected mutants with the highest difference in extent of PC with calcium compared to without calcium. Secondary preference was given to variants that also appeared brighter in the green and red forms. Multiple beneficial amino acid substitutions were combined in small libraries and additional screening and selection was conducted in the same way. Details of the in vitro characterization of CaMPARI variants are provided in Supplementary Methods.

**PC and electrophysiology in rat slice cultures**. Rats were housed and bred at the University Medical Center Hamburg animal facility. All procedures were performed in compliance with German law and according to protocols approved by the Behörde für Gesundheit und Verbraucherschutz of the City of Hamburg.

CaMPARI1 (Addgene #604021) or CaMPARI2 (without C-terminal epitope tags) were subcloned into a pAAV vector under the control of a human synapsin1 promoter. Mutations F391W and L398V were introduced in CaMPARI2, resulting in CaMPARI2_notags_F391W-L398V.

Hippocampal slice cultures from Wistar rats were prepared at postnatal days 4–7 as described[19]. No antibiotics were added to the culture medium. At days in vitro (DIV) 13–17, single-cell electroporation was used to transfect CA1 pyramidal neurons with CaMPARI1 or CaMPARI2_notags_F391W-L398V. Thin-walled pipettes were filled with intracellular K-gluconate based solution into which plasmid DNA was diluted to 20 ng $\mu l^{-1}$. The intracellular solution contained in (mM): 135 K-gluconate, 4 $MgCl_2$, 4 $Na_2$-ATP, 0.4 Na-GTP, 10 $Na_2$-phosphocreatine, 3 ascorbate, 0.02 Alexa Fluor 594, and 10 HEPES (pH 7.2). Pipettes were positioned against neurons and DNA was ejected using an Axoporator 800A (Molecular Devices) with 50 hyperpolarizing pulses (−12 V, 0.5 ms) at 50 Hz[20].

Experiments were performed 3–4 days after electroporation (DIV 16–20). The custom-built two-photon imaging set-up was based on an Olympus BX51WI microscope equipped with a LUMPlan W-IR2 ×60/0.9 NA objective (Olympus), controlled by the open-source software package ScanImage[21]. Two pulsed Ti: Sapphire lasers (MaiTai DeepSee, Spectra Physics) controlled by electro-optic modulators (350–80, Conoptics) were used to excite CaMPARI green (980 nm) and red species (1040 nm), respectively. z-Stacks of CaMPARI-expressing neurons were acquired by sequentially scanning each z-plane at 980 and 1040 nm. Emitted photons were collected through objective and oil-immersion condenser (1.4 NA, Olympus) with two pairs of photomultiplier tubes (PMTs, H7422P-40, Hamamatsu). 560 DXCR dichroic mirrors and 525/50 and 607/70 emission filters (Chroma Technology) were used to separate green and red fluorescence. Excitation light was blocked by short-pass filters (ET700SP-2P, Chroma).

Hippocampal slice cultures were placed in the recording chamber of the microscope and continuously perfused with artificial cerebrospinal fluid saturated with 95% $O_2$ and 5% $CO_2$ and consisting of (in mM): 119 NaCl, 26.2 $NaHCO_3$, 11 D-glucose, 1 $NaH_2PO_4$, 2.5 KCl, 4 $CaCl_2$, and 4 $MgCl_2$ (pH 7.4, 308 mOsm) at room temperature (21–23 °C). Whole-cell recordings from CaMPARI-expressing CA1 pyramidal cells were made at room temperature (21–23 °C) with a Multiclamp 700B amplifier (Molecular Devices) under the control of Ephus software written in Matlab (The MathWorks)[22]. Patch pipettes with a tip resistance of 3–4 MΩ were filled with (in mM): 135 K-gluconate, 4 $MgCl_2$, 4 $Na_2$-ATP, 0.4 Na-GTP, 10 $Na_2$-phosphocreatine, 3 ascorbate, and 10 HEPES (pH 7.2, 295 mOsm). Series resistance was <20 MΩ. In current clamp mode, somatic current injections of 1 ms duration and 3.5 nA current amplitude triggered back-propagating action potentials (bAPs) at the resting membrane potential.

Two brief ultraviolet (UV) light pulses (395 nm, 100 ms, 16 mW $mm^{-2}$, 0.1 Hz) were delivered through the objective using a Spectra X Light Engine (Lumencor) just before imaging to photoswitch CaMPARI into its bright state[12]. The PMTs were protected by shutters (Uniblitz) during the UV pulses. To compare brightness and PC of CaMPARI variants, CaMPARI-expressing neurons were patched and stimulated to fire 100 bAPs at 100 Hz, while simultaneously delivering PC light (395 nm, 2 s, 16 mW $mm^{-2}$). After PC, another z-stack was taken to quantify the change in green and red fluorescence. This procedure was repeated 5–6 times. In experiments designed to monitor turnover of photoconverted CaMPARI, a monopolar electrode was placed in *stratum radiatum* to strongly stimulate synapses onto CA1 CaMPARI-expressing neurons (100 pulses of 0.2 ms duration, at 100 Hz). PC was achieved by delivery of UV light (395 nm, 2 s, 16 mW $mm^{-2}$)

with a 1 s delay from stimulus onset. Slices were returned to the incubator between imaging sessions and CaMPARI fluorescence was monitored for up to 72 h.

For the CaMPARI2 turnover experiments, CA1 neurons in rat organotypic hippocampal slice cultures were electroporated at DIV 15 with DNA (20 ng $\mu l^{-1}$) encoding CaMPARI2_F391W-L398V (no epitope tags). Four days later, two-photon z-stacks were collected using two MaiTai Deep See lasers. For each z-plane, two frames (images) were acquired, exciting at 980 nm (2–5 mW measured at the back-focal plane of the objective) and 1040 nm (3–8 mW), respectively. The procedure was repeated several times (up to 3 days). Between imaging sessions, slices were put back in the incubator. To increase intracellular calcium in response to synaptic activity, a monopolar electrode was placed in the stratum radiatum and 0.2 ms electrical pulses were delivered 100 times at 100 Hz at an intensity that induced postsynaptic spiking in a neighboring nontransfected CA1 neuron. UV light (395 nm, 16 mW $mm^{-2}$) was applied for 2 s with a 1 s delay relative to the start of stimulation (24 neurons, 4 slices). Images were taken at time points $t = -0.5, 0, 6, 12, 24, 30, 48, 60, 72$, and 72.5 h with $t = 0$ immediately after stimulation +UV light or UV light alone. To compare intensity between imaging sessions, images were normalized to a calibration solution containing 200 $\mu g\ ml^{-1}$ fluorescein and 100 $\mu g\ ml^{-1}$ sulforhodamine 101. These higher dye concentrations were selected to closely match the intensity of the CaMPAR2 green species prior to PC and the red species after PC. Fluorescence values of the CaMPARI green and red channels were divided by the corresponding calibration values. R/G is the normalized red fluorescence divided by the normalized green fluorescence.

A macro written in Fiji[23] was used for image analysis (see Supplementary Methods). Image stacks taken with 2P excitation at 980 nm and 1040 nm were *xyz*-aligned to correct for chromatic aberration[24]. After median filtering and rolling ball background subtraction, fluorescence values were obtained from regions of interest (ROIs) drawn onto maximum intensity projections. As the brightness of CaMPARI versions varies, higher laser power is required to image neurons expressing CaMPARI1 than CaMPARI2_F391W-L398V (no epitope tags) (980 nm: 4–10 vs. 3–5 mW; 1040 nm: 10–15 vs. 4–8 mW, respectively, when measured at the back-focal plane of the objective). To allow comparison between imaging sessions, 1 ml of an aqueous solution containing 2 $\mu g\ ml^{-1}$ fluorescein (Alcon) and 0.2 $\mu g\ ml^{-1}$ sulforhodamine 101 (Tocris) was placed in the imaging chamber after each session and fluorescence intensity was measured with identical acquisition settings used to collect the CaMPARI images in that session. The green and red CaMPARI fluorescence values were normalized by dividing by the values from the calibration solution. The R/G ratio is the normalized red fluorescence divided by the normalized green fluorescence. Fold R/G is $(R/G)_{post}$ divided by $(R/G)_{pre}$.

**CaMPARI2 in larval zebrafish**. All zebrafish experiments were conducted in accordance with the animal research guidelines from the National Institutes of Health and were approved by the Institutional Animal Care and Use Committee and Institutional Biosafety Committee of Janelia Research Campus.

Zebrafish (*Danio rerio, casper* background) embryos in the 1–2 cell stage were injected with Tol2 vector containing CaMPARI2 under control of the *elavl3* pan-neuronal promoter. At 6 days post-fertilization (dpf), fish were screened and founders were selected. Founders were crossed with *casper* background fish and offspring with bright fluorescence in the central nervous system were used.

The freely swimming fish were photoconverted for 30 s with 400 nm LED array (Loctite, 200 mW $cm^{-2}$) in the presence or absence of 0.24 mg $ml^{-1}$ tricaine methanesulfonate (MS-222, Sigma). Control fish were not illuminated. All fish were then transferred to 0.24 mg $ml^{-1}$ tricaine and mounted in 1% agarose for imaging.

The forebrain of zebrafish larvae (white box in Fig. 3) was imaged using a Zeiss 710 confocal microscope using a ×20 water immersion objective. The imaged field of view was 250.1 × 250.1 $\mu m^2$ with a 4.1-μs dwell time. For the green channel, we excited with 5% 488 nm and 0.5% 405 nm light (to compensate for photoswitching effects), detecting at 495–554 nm with a detection gain of 600. For the red channel, the 561 nm laser was set at 7% with and detection range was 566–700 nm at a detection gain of 700. z-Slices were acquired every 10 μm.

We imaged for each of the two transgenic lines, 2 control fish (not photoconverted), 4 (CaMPARI1) or 3 (CaMPARI2) fish that were photoconverted in the presence of 0.24 mg $ml^{-1}$ tricaine and 4 fish each that were photoconverted in the absence of tricaine. Using an in-house developed MATLAB script, individual cells were segmented (between 1800 and 6000 cells per condition) and their red-to-green fluorescence signal was determined.

Note that the CaMPARI1 fish used in this study are Tg(elavl3:CaMPARI (W391F+V398L))[jf9] fish, which we made and used in our previous study[12]. These fish express a CaMPARI1 variant with a $K_d$ for $Ca^{2+}$ of 200 nM.

**CaMPARI2(-F391W) in mouse visual cortex**. All experiments were conducted in accordance with the animal research guidelines from the National Institutes of Health and were approved by the Institutional Animal Care and Use Committee and Institutional Biosafety Committee of Cleveland Clinic Lerner Research Institute. Mice were held in standard housing cages with ad libitum access to food and water. Adult mice were anesthetized using isoflurane (2.5% for induction and 1.5% during surgery) in oxygen and placed onto a heated pad (37 °C). A hole was drilled in the skull and AAV was injected into the cortex carrying the CaMPARI2 or CaMPARI2-F391W sequence under the human synapsin1 promoter. The injections were targeted to the left primary visual cortex (V1) and centered around

coordinates 2.7 mm lateral and 0.2 mm anterior to Lambda (4 injections, 35 nl each of ~1 × 10[12] gc ml[−1] AAV solution). A cranial window (two glued layers of #1 glass, Warner Instruments) and a custom headbar were cemented to the skull. Mice were imaged 15–21 days (CaMPARI2) and 15–67 days (CaMPARI2-F391W) after the AAV injection. Prior to starting the imaging session, mice were injected with chlorprothixene hydrochloride (30 µl of 0.33 mg ml[−1] solution, intramuscular, Santa Cruz) and kept lightly anesthetized during imaging (0.5% isoflurane). PC light was illuminated during presentation of a drifting grating moving upward (north direction) stimulus to the mouse right eye. We used an X-Cite Fire lamp (Excelitas) and 440 nm short-pass filter (Semrock FF01-440/SP) with up to 100 mW illumination in the sample plane. PC light was focused to a 7 mm diameter circle with up to ~260 mW cm[−2] and was shined for total of 40 s, 40 cycles of 1 s illumination and 11 s break for tissue cooling, or 20 cycles of 2 s illumination and 10 s break. Imaging was performed using a Bergamo II two-photon microscope with a resonant scanner at 30 Hz acquisition rate and ThorImage software (Thorlabs) with 512 × 512 pixels covering 200 × 200 µm$^2$ of tissue. The light source for two-photon imaging was Insight X3 (Spectra-Physics). We used 950 nm excitation to image functional changes of the CaMPARI green fluorescence to drifting grating moving in 8 different directions (4 s of drifting grating movement followed by 8 s of blank display), and 1040 nm excitation to acquire red and green images to calculate the R/G ratio (without visual stimulus).

Data analysis was performed in Matlab (Mathworks) similar to previous published works[12,17,25]. Cells bodies were segmented using a semi-automatic algorithm[17], and neuropil contamination was corrected using $r = 0.6$. Only cells with baseline fluorescence >90% of their surrounding neuropil signal were included in the analysis. Responsive cells were identified using an ANOVA test with $p < 0.01$. Cells were identified as PC-tuned if they were both responsive and their response to the northward moving grating were significantly lower than their response to a blank display stimulus. We used the OSI as previously defined to describe the tuning level of individual cells to the drifting grating stimuli[16].

**Anti-CaMPARI-red antibody**. EosFP was purified from *E. coli* and photo-converted using 400 nm LED light (Loctite). The photoconverted red form was selectively precipitated at 69 °C and spun down; the green supernatant was further photoconverted and precipitated as well. The orange colored pellet of photo-converted, precipitated EosFP was resuspended in papain protease solution (Sigma) and digested at 37 °C for 5 h followed by 24 h at room temperature. The orange-brown colored digest supernatant was filtered and loaded on a size-exclusion column (Superdex 200, GE Healthcare) and fractionated by isocratic elution with 1 mM Tris, 5 mM NaCl, pH8. The small chromophore-containing fragment, absorbing at 450 nm, was retained longer on the column than other peptide fragments and was collected and verified by liquid chromatography/mass spectrometry (Agilent) to comprise the red chromophore and the next two amino acids (Asn, Arg). This antigen was used by Genscript (Piscataway, NJ, USA) for monoclonal antibody production in mice following standard protocols. Briefly, the antigen was linked to KLH and five immunizations each were given to Balb/c or C57 mice. After screening mouse bleeds for specific antigen recognition (enzyme-linked immunosorbent assay (ELISA) and western blot), three mice were selected and hybridoma libraries were prepared from each. Supernatants from individual hybridoma clones were screened by ELISA with both positive and negative selection using denatured full-length red and green CaMPARI, respectively. Hits were confirmed via ELISA and western blot for specific recognition of the red but not the green form of CaMPARI. Three promising hybridoma clones were scaled up for monoclonal antibody production. After testing each by western blot and immunofluorescence, the sequence of the variable regions of the best-performing antibody was determined by Genscript using rapid amplification of cDNA ends (Figure S13).

**CaMPARI2 PC and histochemistry in HeLa cells**. Hela cells were transfected with pCAG-CaMAPRI2 and a vector expressing the ATP-gated Ca$^{2+}$ channel P2X using the manufacturer's protocols (Lipofectamine 2000, Invitrogen). After 24 h, cells were washed with Hank's Balanced Salt Solution, and ATP was administered to a final concentration of 100 µM immediately followed by PC of one field of view (20 s of 400-nm light at 3 mW cm$^{−2}$) on a Nikon Ti Eclipse wide-field microscope equipped with an LED illuminator (SPECTRA-X, Lumencor), a ×20 objective and an sCMOS camera (Zyla, Andor). The cells were then fixed using formaldehyde (4% in phosphate-buffered saline (PBS) containing 10 mM EGTA, 10 min) and immunostained using standard protocols with anti-CaMPARI-red (1:10,000) and rabbit anti-FLAG (1:3000, F7425, Sigma-Aldrich) as primary and goat-anti-rabbit Alexa Fluor 405 (1:1000; A31556, Invitrogen) and goat-anti-mouse Alexa Fluor 647 (1:1000, A21236, Invitrogen) as secondary antibodies. Cells were then imaged on the same microscope using the 4,6-diamidino-2-phenylindole, fluorescein isothiocyanate, tetramethylrhodamine, and Cy5 imaging channel. In the resulting four-color image, individual cells were segmented in the green channel and four color intensities were calculated.

**CaMPARI2 PC in mouse cortex and histochemistry**. All experimental procedures were approved by the Institutional Animal Care and Use Committee at

Brandeis University and followed the guidelines of the National Institutes of Health.

Male B6.129-Camk4t$^{m1Gsc}$/Ieg mice (EMMA EM:02126) were subject to stereotactic injection of 100 nl of AAV2/1_hsyn1_CaMPARI2 at post-natal day (p) 15[26], then implanted with a fiberoptic cannula made in house (250-µm core diameter, 0.66 NA, ~0.5 mm length, 1.25 ferrule diameter) at p22. At p36, mice were subject to PC light (~50 mW cm$^{−2}$ 390 nm LED connected to the cannula by a custom fiber with 500-µm core diameter, 0.63 NA, and 0.5 m length (Prizmatix)) for 1 h followed immediately by brain dissection. Coronal slices containing V1 were prepared as previously described[27,28], except at 150-µm thickness. Slices were then placed on a glass slide, coverslipped, and imaged on a Leica SP5 confocal microscope. Image stacks with 488 and 543 nm lasers were obtained using a ×20 objective, an optical section height (z-step) of 2 µm, and at 512 × 512 resolution.

Following this initial imaging session, the outline of each slice was traced onto the back of the microscope slide to allow subsequent re-imaging of the same ROIs. Slices were fixed in 4% paraformaldehyde (PFA) in PBS containing 10 mM EGTA for 1 h. These were rinsed three times in PBS and then incubated in the following solutions (each made in PBS with 10 mM EGTA) for the indicated durations on a rotating shaker at room temperature: permeabilization buffer (0.5% TritonX-100, 20% dimethyl sulfoxide (DMSO), 0.3 M glycine, 2 h), blocking buffer (0.5% TritonX-100, 10% DMSO, 6% Normal Goat Serum (NGS), 1 h), primary antibody buffer (0.5% Tween20, 5% DMSO, 3% NGS, overnight), wash buffer (0.5% Tween20, 3 × 30 min), secondary antibody buffer (0.5% Tween20, 3% NGS, 4 h), and wash buffer (0.5% Tween20, 3 × 30 min). Primary antibodies consisted of rabbit anti-FLAG (1:500, F7425, Sigma-Aldrich) and mouse anti-CaMPARI-red (1:1000). Secondary antibodies consisted of anti-rabbit Alexa Fluor 405 (1:400; A31556, Invitrogen) and anti-mouse Alexa Fluor 647 (1:400; A21236, Invitrogen). Following the final wash, the slices were re-aligned to their previous positions on slides, then mounted with an anti-fade medium (Fluoromount-G; SouthernBiotech), coverslipped, and imaged as before, with the addition of 405 and 647 nm lasers. Images were subsequently analyzed in ImageJ.

**CaMPARI2 in vivo PC and histochemistry in rats**. All experiments followed the guidelines outlined in the Guide for the Care and Use of Laboratory Animals (Eighth edition; http://grants.nih.gov/grants/olaw/Guide-for-the-Care-and-Use-of-Laboratory-Animals.pdf) and were approved by the Institutional Animal Care and Use Committee and Institutional Biosafety Committee of the Intramural Research Program of the National Institute on Drug Abuse.

To demonstrate in vivo labeling of active neurons using CaMPARI2, we photoconverted neurons in the prelimbic cortex of rats during mild foot shock. We used male Long-Evans rats (Charles River Laboratories; total $n = 4$), weighing between 300 and 350 g at the time of surgery. Rats were housed individually before and after surgery under a reverse 12 h light/dark cycle (lights off at 8:00 a.m.). Food and water was available ad libitum throughout the experiment. Rats were injected bilaterally with AAV2/1-hSyn-CaMPARI2 in the prelimbic cortex (anterior–posterior (AP) +3.2, medial–lateral (ML) ±0.7, dorsal–ventral (DV) −3.45) and a 400 µm optical fiber was implanted unilaterally for PC in one hemisphere (AP +3.2, ML ±0.7, DV −3.2). The opposite hemisphere was used as the expression control. PC experiments were performed at least 10 days post-surgery to allow sufficient expression of CaMPARI2 in neurons. Rats were connected to the PC laser using custom fiber-optic patch cables and then housed in behavioral testing chambers overnight to minimize non-specific activation. In the morning, rats were presented with a train of mild foot shocks (0.5 mA shock, 12 shocks, 0.5 s ON, 4.5 s OFF, 60 s total presentation) and a 375 nm UV laser (10 mW, CW) was used to photoconvert neurons activated by this aversive stimulus. Rats were perfused for subsequent immunohistochemistry, 90 min after the first shock pulse.

Immediately following behavioral testing, rats were anesthetized with isofluorane and perfused transcardially with ~250 ml of 1× PBS with 10 mM EGTA at pH 7.4 (PBS-E), followed by ~250 ml of 4% PFA with 10 mM EGTA at pH 7.4 (PFA-E). Brains were extracted, post-fixed in PFA-EGTA for 2 h, and then transferred to 30% sucrose with 10 mM EGTA for 48 h at 4 °C. Equilibrated brains were frozen on dry ice and stored at −80 °C. Coronal sections (40 µm) containing prelimbic cortex were then cut using a Leica cryostat, collected in PBS-EGTA, and stored at 4 °C until further processing. Free-floating sections were first rinsed in PBS-E with 0.5% Tween20 and 10 µg ml$^{−1}$ heparin (wash buffer, 3 × 10 min). Sections were incubated in PBS-E with 0.5% TritonX-100, 20% DMSO, and 23 mg ml$^{−1}$ glycine for 3 h at 37 °C (permeabilization buffer) and then in PBS-E with 0.5% TritonX-100, 10% DMSO, and 6% normal donkey serum (NDS) for 3 h at 37 °C (blocking buffer) prior to antibody labeling. Primary antibody was diluted 1:2000 in PBS-E with 0.5% Tween20, 5% DMSO, 3% NDS, and 10 µg ml$^{−1}$ heparin (1° Ab buffer) and sections were incubated overnight in this solution at 37 °C. Following, 1° antibody labeling, sections were rinsed in wash buffer (3 × 10 min) and then incubated in 2° antibody solution (2° antibody diluted 1:500 in PBS-E with 0.5% Tween20, 3% NDS, and 10 µg ml$^{−1}$ heparin) overnight. Following, 2° antibody labeling, sections were rinsed again in wash buffer (3 × 10 min), mounted onto gelatin-coated slides, partially dried, coverslipped with MOWIOL mounting medium, and allowed to hard-set overnight prior to imaging on a Nikon C2 confocal microscope.

Confocal z-stacks of prelimbic cortex were acquired just below the optical fiber on a Nikon C2 confocal microscope using a ×20/0.75 NA air objective. The imaged field of view was $318.20 \times 318.20$ μm$^2$ (0.3107 μm/pixel) with a 5.3-μs dwell time. For native CaMPARI2 green fluorescence, tissue was excited with 4% laser power at 488 nm and emitted fluorescence was collected from 500 to 550 nm with a detection gain of 75 and an offset of −4. To image immunolabeled CaMPARI2 (epitope tag), tissue was excited with 2% laser power at 640 nm and emitted fluorescence collected from 670 to 1000 nm with a detection gain of 75 and an offset of 3. For native CaMPARI2 red fluorescence tissue was excited with 10% laser power at 561 nm and emitted fluorescence was collected from 570 to 1000 nm with a detection gain of 75 and an offset of 4. To image immunolabeled red CaMPARI2 (anti-red antibody), tissue was excited with 20% laser power at 405 nm and emitted fluorescence was collected from 417 to 477 nm with a detection gain of 100 and an offset of 7. All four channels were imaged sequentially (order: 488, then 567, then 647, and finally 405) to minimize overlap and prevent CaMPARI2 PC during image acquisition from confounding colocalization analysis. z-Slices containing all four channels were acquired every 2 μm and the maximum intensity projection for each channel over z was used for colocalization analysis.

**Reagent availability**. DNA constructs for pAAV_hsyn_CaMPARI2, pAAV_h-syn_CaMPARI2_F391W, pAAV_hsyn_CaMPARI2_H396K, pAAV_hsyn_CaM-PARI2_F391W-G395D, and pAAV_hsyn_CaMPARI2_L398T are available via Addgene (http://www.addgene.org #101060–#101064). AAV virus can be requested at the University of Pennsylvania Vector Core (http://www.med.upenn.edu/gtp/vectorcore). Tg(elavl3:CaMPARI2)$^{jf92}$ transgenic zebrafish are deposited to the ZIRC (https://zebrafish.org). Drosophila expressing CaMPARI2 and CaMPARI2-L398T under UAS and LexA promoter in chromosome 2 (su(Hw)attP5), 3 (VK00005), or X (su(Hw)attP8) are available from the Bloomington Drosophila Stock Center (https://bdsc.indiana.edu, #78316–#78326).

## Data availability
Source data from experiments in this study are available from the authors upon reasonable request.

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

## Acknowledgements
The authors thank Amy Hu, Charles Kim, John Macklin, Sujatha Narayan, Iris Ohmert, Ronak Patel, Brenda Shields, and Deepika Walpita for technical assistance and Erik Snapp and Luke Lavis for feedback on the manuscript. This work was supported by the NeuroCure Cluster of Excellence Exc257 and the German Research Council (DFG, Deutsche Forschungsgemeinschaft) Collaborative Research Grant SFB665 (TP11) to B.J.E., the European Union's Horizon 2020 research and innovation program and Euratom research and training program 20142018 (grant agreement No. 670118) to M.E.L., the Human Brain Project (EU Grant 720270, HBP SGA1, 'Context-sensitive Multisensory Object Recognition: A Deep Network Model Constrained by Multi-Level, Multi-Species Data') to M.E.L. and R.N.S.S, DFG Grant No. LA 3442/3–1 & Grant No. LA 3442/5–1 to M.E.L. and T.A.Z., the National Institutes of Health NINDS F32NS101832 to N.F.T. and R37NS092635 to G.G.T., DFG FOR2419 and LFF Hamburg to C.E.G. and T.G.O., and DFG SPP1665 and SFB936 to T.G.O. J.L. holds a Charité postdoctoral For-schungsstipendium, B.C.F. holds a DAAD scholarship, and B.M. holds a postdoctoral fellowship from the Research Foundation-Flanders (FWO Vlaanderen).

## Author contributions
B.M., G.H., and E.R.S. performed in vitro screening and characterization. B.M. generated and characterized transgenic zebrafish lines. B.M. and T.A.B. generated and characterized transgenic fly lines. A.P.-A. and B.C.F. performed electrophysiology on rat slices. N.F.T. did in vivo photoconversion and immunohistochemistry in mouse tissue, while R.M. performed in vivo photoconversion and immunohistochemistry in the rat brain. H.D. and A.D. performed the experiments involving functional imaging and photoconversion in mouse visual cortex. J.L. and T.A.Z. performed immunohistochemistry on ex vivo photoconverted mouse slices with R.N.S.S. B.M., G.H., R.M., A.P.-A., B.C.F., N.F.T., A.D., D.P., H.D., and E.R.S. analyzed data. H.D., B.J.E., M.E.L., G.G.T., C.E.G., T.G.O., B.T.H., and E.R.S. supervised the work. The manuscript was written by B.M. and E.R.S. with help from R.M., A.P.-A., N.F.T., H.D., and R.N.S.S. All authors reviewed and approved the manuscript.

## Additional information

**Competing interests:** E.R.S. is an inventor on US patent number 9,518,996 and US patent application 15/335,707, which may cover CaMPARI sequences described in this paper. The remaining authors declare no competing interests.

