## [Peer Review File · Nature Communications]

Reviewers' comments:

Reviewer #1 (Remarks to the Author):

In this study Moeyaert et al, developed and presented the next generation photo convertible neural activity indicator CaMPARI2. Authors did a series of directed mutations to evolve the original CaMPARI1 protein to its brighter version with higher photo conversion rations, with different Calcium binding affinity and kinetics. As a result of this directed evolution CaMPARI2 appears to be more effective in specifically detecting neural activity both in rodent and in zebrafish brain. Excitingly the authors also developed a mono clonal mouse antibody to specifically detect photo-converted Red-CaMPARI2, which is certainly a major step-forward for the effective use of this photo convertible neural activity indicator, in combination with classical tools of immuno histochemistry. I find this study very exciting and suitable for publication in Nature Communications, since these new tools CaMPARI2 and its antibody will likely be used by many people investigating neural circuit function, in freely behaving animals and in brain slices. Moreover, the manuscript is very well written, results are well presented.

I only have few remarks that could improve the use of the present study by the community and compare the effectiveness of CaMPARI2 with other tools.

- 1) Red CaMPARI results are presented only for rodent slices. Does this antibody also work well in zebrafish brain? Such a small addition to the manuscript displaying the effectiveness of Red CaMPARI antibody in zebrafish can help the people working with zebrafish.
- 2) Now that the authors present this very effective tool to be used also in fixed samples, it would be interesting to bench mark the effectiveness of photo-converted and immune stained red-CaMPARI2 to classical markers such as cFOS and/or pERK

Reviewer #2 (Remarks to the Author):

This work improves an already established tool (CaMPARI1) that enables marking of active neuronal populations. The previously published genetic tool (CaMPARI1) is a photoconvertible fluorescent protein, which depends on the presence of light and high concentration of free Ca^{+2} ions to switch its green color into red. Thus, red cells represent active cells. In this paper, the authors developed a modified and better version of the cellular sensor (CaMPARI2). The improvements include: 1. increased contrast of green-to-red ratio; 2. higher brightness of both green and red colors; 3. higher rate of calcium unbinding; and 4. decreased background photoconversion in low concentrations of Ca^{+2} . In addition, a monoclonal antibody that specifically recognizes the photoconverted red protein, was developed. This new tool was validated in vitro and in zebrafish larvae.

Major comments

1. This is a pure method paper. There is no attempt to answer a biological question. The methods developed improve several technical drawbacks of the previous tool. The work does not provide major and significant breakthrough.
2. Throughout the paper the authors suggest that their new tool will help to understand the mechanisms of memory and learning. Future work may confirm this assumption; however, the tool can be widely used to elucidate many questions in neuroscience. Why do the authors focus on memory and learning?
3. The tool was generally studied in neurons. It would have been helpful to test it in other cell types, i.e. glial cells and myocytes. This could have been used as a control for the neuron experiments and could extend the use of this technique.
4. It would have been helpful to know whether the tool work differently in excitatory versus inhibitory cells. Post staining with antibodies against markers for GABA and glutamate cells is needed. In the fish experiments, in addition to the pan neuronal promoter, it is recommended to use cell-specific promoters that drive expression in well characterized cell types.

5. In the fish experiments, the number of fish tested is not sufficient. 2-4 larvae is not enough, considering the variability in brain activity of sedated and freely behaving animals. The high-throughput nature of this model should enable experiments with higher n.
 6. Comparison between sedated and freely behaving animal is not adequate. Causal controlled experiments, using manipulation in a specific circuit (silencing and/or activation) should be done.
 7. The advantage of the current method over combined use of IEG and GCaMP should be better explained in the discussion. How much better is this tool than other published tools described in papers like Randlett et al. Nat Methods, 2015; Yang et al. Nat Commun, 2018; and others.
- Minor comments
8. "The c-terminal epitope tags led to increase Ca²⁺ affinity", why?
 9. In the intro: "using directed evolution methods", what does this mean?
 10. "Chemical tissue fixation using formaldehyde, for example, generally results in loss of fluorescent signal due to protein denaturation". "Change in protein conformation" may be more appropriate.
 11. In the figures, the fonts are too small on the y and x axis (for example, in figure 1).

Reviewer #3 (Remarks to the Author):

This manuscript describes the engineering and characterization of CAMPARI2, an improved version of a calcium-dependent green-to-red photoconvertible protein that can be used to mark active neurons. This work addresses three limitations of the original CAMPARI1: background photoconversion in low calcium, slow calcium unbinding kinetics, and the loss of contrast upon fixation. Through systematic brute force engineering, the authors identified improved variants with decreased background photoconversion in low calcium leading to enhanced photoconversion contrast in low versus high calcium. The experiments described in the paper are thorough and rigorous, and I have very little to criticize. The improved CAMPARI2 and the antibody against the red photoconverted chromophore are likely to be highly valuable tools and widely used by the scientific community.

Minor issues:

- 1) Page 2, bottom, authors state: screens were performed "largely as described previously". Please clarify differences between this screen and previous screens.
- 2) Page 3 bottom, authors state: red signal is "largely turned over at 72 hrs". Again, 'largely' is vague and subjective. Please express more quantitatively.
- 3) Caption of Figure S8 states that fluorescence intensities and R/G ratio were normalized to laser power. Please provide more details. Were different laser powers used for different experiments or just for red versus green. Please provide enough information on how data were normalized for someone else to be able to process data in exactly the same way.
- 4) Figure 4 was weak and unconvincing. The data presented in S15B are much stronger and more quantitative. Please consider moving S15B to main manuscript.
- 5) Figure 1, bottom, middle graph is not described in the caption.
- 6) Figure 2: please include x-axis for right hand plots. Again, please detail laser powers used and normalization procedure. Y-axis is labeled "fluorescence (AU)" or "fold R/G" but caption implies these have been normalized. Please clarify.

We thank the reviewers and editors for their comments and suggestions. We have addressed each point raised by the reviewers in the documents below, in the order they were raised. Specifically, each reviewer's comment/suggestion is in bold, our response to that comment/suggestion is in italics, and the changes to the manuscript text or figures is described, with differences highlighted in bold blue text. Additionally, changes to the manuscript text can be viewed using track changes.

In addition to editing and re-arranging parts of the manuscript, we have added additional data demonstrating the use of CaMPARI2 to mark activity in response to specific stimuli *in vivo* in mouse visual cortex. We feel this manuscript was substantially improved as a result of the review and we look forward to your consideration of the revised version.

Reviewer #1:

In this study Moeyaert et al, developed and presented the next generation photo convertible neural activity indicator CaMPARI2. Authors did a series of directed mutations to evolve the original CaMPARI1 protein to its brighter version with higher photo conversion ratios, with different Calcium binding affinity and kinetics. As a result of this directed evolution CaMPARI2 appears to be more effective in specifically detecting neural activity both in rodent and in zebrafish brain. Excitingly the authors also developed a mono clonal mouse antibody to specifically detect photo-converted Red-CaMPARI2, which is certainly a major step-forward for the effective use of this photo convertible neural activity indicator, in combination with classical tools of immuno histochemistry. I find this study very exciting and suitable for publication in Nature Communications, since these new tools CaMPARI2 and its antibody will likely be used by many people investigating neural circuit function, in freely behaving animals and in brain slices. Moreover, the manuscript is very well written, results are well presented

I only have few remarks that could improve the use of the present study by the community and compare the effectiveness of CaMPARI2 with other tools.

1) Red CaMPARI results are presented only for rodent slices. Does this antibody also work well in zebrafish brain? Such a small addition to the manuscript displaying the effectiveness of Red CaMPARI antibody in zebrafish can help the people working with zebrafish.

We initially developed 3 mouse monoclonal antibodies and tried all three of them using several protocols to stain dissected larval zebrafish brains. Although we achieved good staining using an anti-5HT antibody as positive control, we have not been able to achieve efficient and specific labeling using any of our antibodies in zebrafish brain tissue. We have added a short remark in the manuscript that this has been unsuccessful so far. It is currently unclear if our antibodies are somehow incompatible with fish brain tissue (even though they work well in mouse and rat brain tissue), or if we are simply doing something wrong (we do not have much experience with zebrafish immunohistochemistry).

Submitted:

In both cultured cells and rodent brain tissue, the anti-CaMPARI-red antibody specifically recognizes the photoconverted form of CaMPARI and the stained images are reflective of the endogenous red CaMPARI2 signal *in vivo* (Figure 5B, 6 and S16 and S17).

Revised:

In both cultured cells and rodent brain tissue, the anti-CaMPARI-red antibody specifically recognizes the photoconverted form of CaMPARI and the stained images are reflective of the

endogenous red CaMPARI2 signal *in vivo* (Figure 5B, 6 and S16 and S17), **although staining of dissected larval zebrafish brains with anti-CaMPARI-red antibody was unsuccessful in our hands.**

2) Now that the authors present this very effective tool to be used also in fixed samples, it would be interesting to bench mark the effectiveness of photo-converted and immune stained red-CaMPARI2 to classical markers such as cFOS and/or pERK

In our previous work reporting CaMPARI1, we did indeed show a correlation between CaMPARI photoconversion and nuclear phosphorylated CREB in cultured neurons stimulated with high potassium (10.1126/science.1260922, Fig. 1J-L). Similarly, it should be possible to look for correlations between CaMPARI photoconversion and other markers, as the reviewer mentions.

. We therefore believe that a detailed treatment of this subject is beyond the scope of this manuscript. However, we are enthusiastic that CaMPARI2 may help enable this type of work in the future.

Reviewer #2:

This work improves an already established tool (CaMPARI1) that enables marking of active neuronal populations. The previously published genetic tool (CaMPARI1) is a photoconvertible fluorescent protein, which depends on the presence of light and high concentration of free Ca²⁺ ions to switch its green color into red. Thus, red cells represent active cells. In this paper, the authors developed a modified and better version of the cellular sensor (CaMPARI2). The improvements include: 1. increased contrast of green-to-red ratio; 2. higher brightness of both green and red colors; 3. higher rate of calcium unbinding; and 4. decreased background photoconversion in low concentrations of Ca²⁺. In addition, a monoclonal antibody that specifically recognizes the photoconverted red protein, was developed. This new tool was validated *in vitro* and in zebrafish larvae.

Major comments

1) This is a pure method paper. There is no attempt to answer a biological question. The methods developed improve several technical drawbacks of the previous tool. The work does not provide major and significant breakthrough.

The authors acknowledge that this manuscript is methodological in nature and does not bring novel biological insights. However, we are confident that the better performance of CaMPARI2 relative to the original CaMPARI, as well as the possibility of immunohistochemical detection of the photoconverted form will be beneficial to the community and enable novel biological insights. It is in this light that we argue that publication in Nature Communications, which is high-impact and broad in scope and readership, is appropriate.

2) Throughout the paper the authors suggest that their new tool will help to understand the mechanisms of memory and learning. Future work may confirm this assumption; however, the tool can be widely used to elucidate many questions in neuroscience. Why do the authors focus on memory and learning?

Learning and memory are indeed emphasized in both the abstract and the introduction. This primarily reflects the interests of various authors on this work. We do recognize and intended that CaMPARI could be a general tool for marking calcium signaling, which is widespread in neuroscience and biology in general. We have therefore edited the abstract and introduction to be more general.

Submitted:

1. Marking functionally distinct neuronal ensembles with high spatiotemporal resolution is a key challenge in systems neuroscience and critical for understanding the mechanisms of learning and memory.
2. Understanding how memories are formed and reactivated to mediate highly specific learned behaviors is one of the most fundamental challenges in neuroscience. It is thought that specific patterns of sparsely distributed neurons, called neuronal ensembles, are selected by specific stimuli during learning and memory reactivation. Long-lasting alterations or physical traces (often called engrams) are thought to be induced specifically within these neurons to encode long-lasting memories of the experience.¹⁻⁵ Since ensembles are composed of multiple cellular phenotypes,⁶⁻⁹ these neurons can be identified only by their activation state during learning or reactivation of the memory.
3. We have developed CaMPARI2, with higher molecular brightness, faster calcium unbinding kinetics and less background photoconversion in the absence of calcium compared to CaMPARI1, leading to a higher contrast between high-calcium and low-calcium cells.

Revised:

1. Marking functionally distinct neuronal ensembles with high spatiotemporal resolution is a key challenge in systems neuroscience and critical **for obtaining a deeper understanding of neural networks that form the basis of learning, memory, action selection, and other critical behaviors.**
2. **The coordinated activity of neurons that are spatially distributed throughout complex tissues like the brain are thought to mediate critical functions such as the selection and generation of actions in response to stimuli, learning from the outcomes of those actions, and the storage and recall of memories of those actions and outcomes. Methods to identify these neuronal ensembles based on their activity over various time and spatial scales are critical to furthering our understanding of brain function.**
3. **As a complement to existing techniques for marking active neuronal populations, we recently introduced CaMPARI, a fluorescent protein whose green-to-red photoconversion is calcium-dependent. CaMPARI allows marking of active neurons with finer time resolution than activity-dependent gene expression and provides a more permanent signal than transient calcium indicators like GCaMP. In this work, we** have developed CaMPARI2, with higher molecular brightness, faster calcium unbinding kinetics and less background photoconversion in the absence of calcium compared to CaMPARI1, leading to a higher contrast between high-calcium and low-calcium cells.

3) The tool was generally studied in neurons. It would have been helpful to test it in other cell types, i.e. glial cells and myocytes. This could have been used as a control for the neuron experiments and could extend the use of this technique.

Our bias towards use of CaMPARI in neurons again reflects the interests and expertise of the authors. We fully expect that other groups will adopt CaMPARI2 to study calcium signaling in other cell types. We characterized a series of CaMPARI2 variants with a range of affinities for calcium (Figure 1, S4 and S6) specifically to accommodate use in different cell types where calcium levels may be different.

In this work, we do characterize CaMPARI and the anti-red antibody in HeLa cells in addition to neurons.

The HeLa cell data have been moved to main Figure 6 in response to this and another reviewer comments. Methods for the HeLa cell experiment have been added.

Submitted:

1. We identified variants of CaMPARI2 with higher calcium affinity and slower release kinetics (Table S4) that might be preferable if high time resolution is not required.

Revised:

1. We identified variants of CaMPARI2 with higher **or lower** calcium affinity and **variable** calcium release kinetics (Table S4) that might be preferable **in other tissues or cell types**.
2. **CaMPARI2 photoconversion and immunohistochemistry in HeLa cells**
HeLa cells were transfected with pCAG-CaMAPRI2 and a vector expressing the ATP-gated Ca²⁺ channel P2X using manufacturer's protocols (Lipofectamine 2000, Invitrogen). After 24h, cells were washed with HBSS, and ATP was administered to a final concentration of 100 uM immediately followed by photoconversion of one field of view (20 s of 400-nm light at 3 mW/cm²) on a Nikon Ti Eclipse wide-field microscope equipped with a LED illuminator (SPECTRA-X, Lumencor), a 20× objective and an sCMOS camera (Zyla, Andor). The cells were then fixed using formaldehyde (4% in PBS containing 10 mM EGTA, 10 min) and immunostained using standard protocols with anti-CaMPARI-red (1:10 000) and rabbit anti-FLAG (1:3000, F7425, Sigma-Aldrich) as primary and goat-anti-rabbit Alexa Fluor 405 (1:1000; A31556, Invitrogen) and goat-anti-mouse Alexa Fluor 647 (1:1000, A21236, Invitrogen) as secondary antibodies. Cells were then imaged on the same microscope using the DAPI, FITC, TRITC and Cy5 imaging channel. In the resulting 4-color image, individual cells were segmented in the green channel and 4 color intensities calculated.

4) It would have been helpful to know whether the tool work differently in excitatory versus inhibitory cells. Post staining with antibodies against markers for GABA and glutamate cells is needed. In the fish experiments, in addition to the pan neuronal promoter, it is recommended to use cell-specific promoters that drive expression in well characterized cell types.

Again, we recognize that CaMPARI2 will likely be used in cell types other than those we show here, and for that reason we describe a panel of calcium affinity variants that should cover a range of calcium levels accessed in many cell types. We agree that the use of cell-type specific promoters can be valuable to limit expression of transgenes to sub-populations of cells for certain types of biological experiments. CaMPARI2 is fully compatible with cell-type specific promoter driven expression, as we show with AAV-synapsin_promoter-CaMPARI2 and HuC (elavl3) promoter to drive expression selectively in neurons for some of our experiments.

(See also the changes made in response to reviewer #2, comment 3)

5) In the fish experiments, the number of fish tested is not sufficient. 2-4 larvae is not enough, considering the variability in brain activity of sedated and freely behaving animals. The high-throughput nature of this model should enable experiments with higher n.

Although a high number of fish can very easily be photoconverted, the imaging and data analysis of this experiment is not so high-throughput. Fish have to be well mounted individually and imaged as a Z-stack on a confocal microscope. Next, cells are segmented by manually selecting a seed pixel in the nucleus, after which an algorithm determines a presumed cell cytosol. Doing this for one fish is quite

laborious, but necessary to get reliable data. For example, as mentioned in the caption of Figure 2, we segmented hundreds to thousands of cells per fish for this work. These considerations of course reduce the throughput of the experiment significantly. We have thought of other options (such as co-staining nuclei with Hoechst to do fully automated segmentation, co-imaging both color channels to speed up imaging), but none gave data we were confident about.

Given this, we included 2 fish for the “not photoconverted” setting and 4 fish for the others (only 3 anaesthetized CaMPARI2 fish) and show data for each fish individually. The purpose of these experiments was to demonstrate two points.

- First, we wanted to show that there is a difference in photoconversion between fish that were both swimming and irradiated and fish that were either sedated or not irradiated. From Figs. 3 and S10, it is immediately clear that this is the case, even with the number of fish presented here.
- Secondly, we wanted to show that CaMPARI2 results in a larger spread of R/G ratio (due to the higher brightness of the probe and the lower background photoconversion) relative to CaMPARI1. This point might not be immediately clear from Figs. 3 and S10, but can be seen in Figure S9. The reason why this could be considered unconvincing is because the cells’ activity is highly variable within one animal. In fact, the variability of the R/G ratios within one animal (SD of .16/.14/.14/.15 (CaMPARI1) and .23/.26/.35/.24 (CaMPARI2)) is much larger than the spread of the R/G ratio means across animals (SD of .0082 (CaMPARI1) and .051 (CaMPARI2)). In other words, including more animals will not bring more information needed to demonstrate this point.

We chose to only include 2 animals in the non-irradiated case, since the red signal measured here is merely reflective of the background of our microscope.

In order to convey these points more clearly to the readership, we included all datapoints on the whiskerplot in Figure S10 and re-phrased the conclusions drawn from Figure 3 and S10.

Submitted:

1. Finally, we compared the CaMPARI photoconversion signal in larval transgenic zebrafish expressing CaMPARI1 and CaMPARI2 under the control of a neuron-specific promoter. When we blocked neuronal activity with the sodium channel blocker tricaine during a photoconversion light pulse, less photoconversion was observed with CaMPARI2 (Figure 3, Figure S9), in line with the lower background photoconversion rates of CaMPARI2 from *in vitro* measurements (Table 1). Photoconversion of freely-swimming fish resulted in higher red/green ratios and a larger range of red-to-green ratios (Figure S9 and S10), demonstrating that it is possible to mark a wider range of activity patterns with CaMPARI2 *in vivo*.
2. Our experiments in larval zebrafish brain showed a similar amount of red fluorescence with CaMPARI1 and CaMPARI2, but with a wider distribution of red/green values with CaMPARI2, indicating that it can delineate finer gradations of activity.

Revised:

1. **The photoconversion signal was then compared *in vivo* in larval transgenic zebrafish expressing either CaMPARI1 or CaMPARI2 from a neuron-specific promoter. When neuronal activity was blocked with the sodium channel blocker tricaine during a photoconversion light pulse, the red-to-green ratios were slightly lower with CaMPARI2, in agreement with a lower baseline photoconversion rate for CaMPARI2 (Figure 3, Figures S9 and S10). After photoconversion of freely-swimming fish to mark neurons with ongoing spontaneous activity, we measured many**

neurons with higher red/green ratios and a larger range of red/green ratios with CaMPARI2 (Figures S9 and S10), suggesting that it is possible to mark activity on a fine scale in the CaMPARI2 transgenic zebrafish.

2. Our experiments in larval zebrafish brain showed a similar amount of red fluorescence with CaMPARI1 and CaMPARI2, and a wide distribution of red/green values among CaMPARI2-expressing neurons, indicating that it can delineate fine gradations of activity.

6) Comparison between sedated and freely behaving animal is not adequate. Causal controlled experiments, using manipulation in a specific circuit (silencing and/or activation) should be done.

We agree with the reviewer that characterization of CaMPARI response to controlled stimuli in characterized circuits or pathways is important. For that reason, we performed such experiments in several model organisms in our previous publication (Fosque et al., Science 2015) using CaMPARI1. This current manuscript introduces an improved version of CaMPARI that is conceptually identical, and our careful in vitro measurements illustrate exactly how various biophysical parameters (brightness, contrast, affinity for calcium, sensitivity to light, kinetics, etc.) have been altered. We believe that repeating all of these will not add value to the current manuscript.

However, we have coordinated with one co-author (Hod Dana, now running his own lab at the Cleveland Clinic) to carry out experiments in mouse primary visual cortex in response to controlled drifting grating stimuli, as was done in the original CaMPARI publication. These results show that CaMPARI2 and a higher-affinity variant (CaMPARI2-F391W) both allow marking of neuron populations that show specific responses to visual stimuli. The data is presented in the main text, accompanied by Figure 4 and S11.

Submitted:

1. We demonstrate the improved performance of CaMPARI2 in mammalian neurons and *in vivo* in larval zebrafish brain.

Revised:

1. We demonstrate the improved performance of CaMPARI2 in mammalian neurons and *in vivo* in larval zebrafish brain and mouse visual cortex.
2. Finally, to demonstrate the ability of CaMPARI2 to mark neurons in response to specific stimuli *in vivo*, we expressed CaMPARI2 and CaMPARI2-F391W in the mouse visual cortex and measured CaMPARI2 photoconversion as well as traditional calcium indicator fluorescence in response to specific visual stimuli. Bright CaMPARI2 labeling was evident 15 days after AAV injection and individual layer 2/3 neurons could be easily identified (Figure 4A). Lightly anesthetized mice were presented with upward drifting gratings while illuminated with photoconversion light. Following the photoconversion, the calcium response of individual neurons was recorded in response to the presentation of drifting gratings in eight different directions (Figure 4B, Methods).

We identified neurons with calcium responses (ANOVA test, $p < 0.01$) during presentation of any of the visual stimuli and grouped them as neurons with a significant calcium response during the upward drifting gratings displayed during photoconversion (PC-tuned) and with no significant response to the upward drifting grating but with significant responses to other directions of motion (responsive but not PC-tuned). In addition, we calculated the OSI of each neuron and grouped together all neurons that were responsive but had $OSI < 0.5$, suggesting they were broadly responsive to all directions of motion and likely inhibitory (broadly-tuned, Figure 4C). It was previously shown that inhibitory neurons have lower response amplitudes

during calcium imaging²⁸ and lower orientation selectivity index (OSI²⁶) for responding to drifting grating stimuli than excitatory neurons^{26,27}, allowing us to categorize them separately based on their response profile.

Of all segmented neurons, 11.5% were identified as responsive (48/359, 18/203 and 41/525, 48/366, 63/419 for two CaMPARI2 and three CaMPARI2-F391W mice respectively) and 3.2% was PC-tuned (10, 1 and 17, 26, 6 for two CaMPARI2 and three CaMPARI2-F391W mice respectively). The red-to-green ratio of the responsive cells was correlated to the peak change in fluorescence for the northward moving grating stimulus (Figure 4C, S11 top), but not to the orthogonal direction stimulus (Figure S11 middle). The red-to-green ratio was significantly higher for PC-tuned cells than both responsive and not PC-tuned cells and cells that had no significant change to the visual stimuli (Figure 4D, S11 bottom, Wilcoxon Rank Sum Test), demonstrating that CaMPARI2 can mark neurons that are responsive to specific stimuli *in vivo*. We note that the photoconversion of CaMPARI2 yielded better separation among these groups than CaMPARI2-F391W (Figure S11), presumably because the higher calcium affinity of CaMPARI2-F391W led to partial saturation in the absence of stimuli which resulted in higher baseline photoconversion and lower contrast between responsive and non-responsive neurons.

3. *In vivo* calcium imaging and photoconversion in mouse visual cortex confirms that CaMPARI2 is able to selectively label neurons that are active during a specific stimulus.

7) The advantage of the current method over combined use of IEG and GCaMP should be better explained in the discussion. How much better is this tool than other published tools described in papers like Randler et al. Nat Methods, 2015; Yang et al. Nat Commun, 2018; and others.

We agree that understanding the relative pros and cons is important to guide and facilitate the use of any tool. We discuss at some length the relative advantages and disadvantages of IEG- and GCaMP-based methods in the first paragraph of the introduction. We do not claim that CaMPARI is better than previously-published cell marking tools since “better” is relative and depends highly on biological question being asked. Since the focus of this paper is the improved performance of CaMPARI2 rather than the concept of a CaMPARI-like neuron marking approach, we chose not to focus on those points in the discussion. However, we have added a sentence to the beginning of the discussion summarizing this briefly.

Submitted:

We have developed CaMPARI2, with higher molecular brightness, faster calcium unbinding kinetics and less background photoconversion in the absence of calcium compared to CaMPARI1, leading to a higher contrast between high-calcium and low-calcium cells.

Revised:

As a complement to existing techniques for marking active neuronal populations, we recently introduced CaMPARI, a fluorescent protein whose green-to-red photoconversion is calcium-dependent. CaMPARI allows marking of active neurons with finer time resolution than activity-dependent gene expression and provides a more permanent signal than transient calcium indicators like GCaMP. In this work, we have developed CaMPARI2, with higher molecular brightness, faster calcium unbinding kinetics and less background photoconversion in the absence of calcium compared to CaMPARI1, leading to a higher contrast between high-calcium and low-calcium cells.

Minor comments

8) “The c-terminal epitope tags led to increase Ca²⁺ affinity”, why?

Short answer: we don't know. The epitope tag is C-terminally fused, so in primary and most likely also tertiary space it is very close to the calmodulin-binding peptide and the calmodulin moiety. As can be seen from Figure S1, the epitope tags contain several negatively charged amino acids that might interact with the positively charged residues in the calmodulin-binding domain. However, the authors have no evidence of how this would lead to an increase in Ca²⁺-affinity. Many examples exist in the literature of terminal tags altering protein function and it is generally assumed to occur through interactions between the protein of interest and the tags or between the tags and other biomolecules in the cell. We chose to leave this particular observation described (we show the altered calcium affinity) but unexplained. A systematic mutagenesis effort might shed additional light on why these tags alter calcium affinity, but honestly does not seem worthwhile. We added a note mentioning that we could not find an obvious explanation for this observation.

Submitted:

Addition of the C-terminal epitope tags led to increased calcium affinity (Figure S2).

Revised:

Addition of the C-terminal epitope tags led to increased calcium affinity (Figure S2), **although the reasons for this observation are unclear.**

9) In the intro: “using directed evolution methods”, what does this mean?

Although this is a methods paper, we chose to focus on the results of our protein engineering and how they can benefit the community rather than on the technicality of the protein engineering. As such, we did not elaborate on the materials and methods used for the directed evolution. The term “directed evolution” is first introduced in the introduction, where no further information is given for brevity and conciseness. In the results section, the second sentence gives more detail on how we approached the directed evolution. The first paragraph of the Materials and Methods section also gives some technical background.

However, we acknowledge that this term might be vague for at least a part of the readership, and made textual changes in order to better explain how CaMPARI2 was designed.

(See also the changes made in response to reviewer #3, comment 1)

Submitted:

1. In this work we present CaMPARI2, an improved version of CaMPARI1. Using directed evolution methods, we significantly increased the contrast of green-to-red photoconversion between the calcium-bound and calcium-free states.
2. We targeted mutagenesis to amino acid positions around the fluorescent protein chromophore and at the protein interface between the fluorescent protein and the calcium-sensitive domains of CaMPARI, as defined by the crystal structures of CaMPARI and other GECIs.^{20,22,23} We assayed fluorescence and photoconversion contrast (extent of photoconversion in high-calcium versus low-calcium conditions) of ~950 unique single amino acid substitutions at 50 separate positions of CaMPARI1_W391F-V398L in a medium-throughput assay in bacterial lysates, largely as described previously.²⁰

Revised:

1. In this work we present CaMPARI2, an improved **variant of CaMPARI1. Using site-directed amino acid mutagenesis combined with functional screening and selection**, we significantly increased the contrast of green-to-red photoconversion between the calcium-bound and calcium-free states.
2. We targeted **site-saturation** mutagenesis to amino acid positions around the fluorescent protein chromophore and at the protein interface between the fluorescent protein and the calcium-sensitive domains of CaMPARI, as defined by the crystal structures of CaMPARI and other GECIs.^{20,24,25} We assayed fluorescence and photoconversion contrast (extent of photoconversion in high-calcium versus low-calcium conditions) of ~950 unique single amino acid substitutions at 50 separate positions of CaMPARI1_W391F-V398L in a medium-throughput assay in bacterial lysates, **as described** previously.²⁰

10) “Chemical tissue fixation using formaldehyde, for example, generally results in loss of fluorescent signal due to protein denaturation”. “Change in protein confirmation” may be more appropriate.

We agree with the reviewer that “change in protein conformation” describes exactly what we believe is happening (more general than “protein denaturation”) and we have changed the text to reflect this wording.

Submitted:

Although the red form of CaMPARI2 is bright and easily detectable in live cells and tissue with conventional fluorescence microscopy, chemical tissue fixation using formaldehyde, for example, generally results in loss of fluorescent signal due to protein denaturation.

Revised:

Although the red form of CaMPARI2 is bright and easily detectable in live cells and tissue with conventional fluorescence microscopy, chemical tissue fixation using formaldehyde, for example, generally results in loss of fluorescent signal due to **changes in protein conformation following chemical modification by formaldehyde**.

11) In the figures, the fonts are too small on the y and x axis (for example, in figure 1).

We agree and increased the font size of:

- *Figure 1, panel B and C, axes and legends*
- *Figure 2, most text. Additionally, the font has been changed to Arial for uniformity among all main figures. Other changes have been made to this figure in response to remarks made by reviewer #3.*
- *Figure 4 (submitted) / Figure 5 (reviewed), all text*

Reviewer #3 (Remarks to the Author):

This manuscript describes the engineering and characterization of CaMPARI2, an improved version of a calcium-dependent green-to-red photoconvertible protein that can be used to mark active neurons. This work addresses three limitations of the original CaMPARI1: background photoconversion in low calcium, slow calcium unbinding kinetics, and the loss of contrast upon fixation. Through systematic brute force engineering, the authors identified improved variants with decreased background photoconversion in low calcium leading to enhanced photoconversion contrast in low versus high calcium. The experiments described in the paper are thorough and rigorous, and I have very little to criticize. The improved

CAMPARI2 and the antibody against the red photoconverted chromophore are likely to be highly valuable tools and widely used by the scientific community.

Minor issues:

1) Page 2, bottom, authors state: screens were performed “largely as described previously”. Please clarify differences between this screen and previous screens.

This phrasing is indeed not specific enough. We have reworked the methods section to include more details about the screening procedure. Also, since the screening method itself was identical to the one previously used (only the templates were different), we removed the word “largely”.

Submitted:

1. For the directed evolution of CaMPARI, green and red fluorescence of bacterial lysates in 0.5 mM CaCl₂ or 1 mM EGTA was acquired using a fluorescence plate reader. Fluorescence was measured again after irradiation with 405 nm light, and again after addition of 10 mM EGTA and 5 mM CaCl₂, respectively. From these fluorescence reads, we selected mutants with the highest difference in extent of photoconversion with vs. without calcium. Secondary preference was given to variants that also appeared brighter in the green and red forms. Additional details of the development and *in vitro* characterization of CaMPARI variants are provide in the supplemental methods.
2. We assayed fluorescence and photoconversion contrast (extent of photoconversion in high-calcium versus low-calcium conditions) of ~950 unique single amino acid substitutions at 50 separate positions of CaMPARI1_W391F-V398L in a medium-throughput assay in bacterial lysates, largely as described previously.

Revised:

1. **We conducted multiple rounds of site-saturation mutagenesis, functional screening and selection to improve the properties of CaMPARI. Site-saturation mutagenesis at individual amino acid positions was done using the QuikChange Multi protocol (Agilent). Generally, 90 colonies were picked from a site-saturation mutagenesis library at each amino acid position, along with controls, into deep-well 96-well blocks. The T7 Express *E. coli* bacteria (New England Biolabs) were grown at 30 °C for 36 h and pelleted by centrifugation. Soluble lysate was prepared from the pellets by incubation with Bacterial Protein Extraction Reagent (Thermo Fisher) followed by centrifugation. Functional screening included measurement of green and red fluorescence of bacterial lysates using a fluorescence plate reader (Tecan) after addition of 0.5 mM CaCl₂ or 1 mM EGTA to separate lysate aliquots. Fluorescence was measured again after irradiation with 405 nm light using an LED array (Loctite; 1 min, ~200 mW/cm²), and again after addition of 10 mM EGTA and 5 mM CaCl₂, respectively. From these fluorescence reads, we selected mutants with the highest difference in extent of photoconversion with calcium compared to without calcium. Secondary preference was given to variants that also appeared brighter in the green and red forms. Multiple beneficial amino acid substitutions were combined in small libraries and additional screening and selection was conducted in the same way. Details of the *in vitro* characterization of CaMPARI variants are provided in the Supplemental Methods.**
2. We assayed fluorescence and photoconversion contrast (extent of photoconversion in high-calcium versus low-calcium conditions) of ~950 unique single amino acid substitutions at 50 separate positions of CaMPARI1_W391F-V398L in a medium-throughput assay in bacterial lysates, **as described previously.**

2) Page 3 bottom, authors state: red signal is “largely turned over at 72 hrs”. Again, ‘largely’ is vague and subjective. Please express more quantitatively.

We agree that this is a vague term and have updated this sentence to a more quantitative statement.

Submitted:

After photoconverting during synaptic stimulation, the red signal persists for at least 24 h, but is largely turned over at 72 h allowing re-labelling by repeating the stimulation and photoconversion (Figure S8).

Revised:

After photoconverting during synaptic stimulation, the red signal **is strongly visible for at least 24 h, but the R/G ratio falls to ~20% of its maximum value after 72 h**, allowing re-labelling by repeating the stimulation and photoconversion (Figure S8).

3) Caption of Figure S8 states that fluorescence intensities and R/G ratio were normalized to laser power. Please provide more details. Were different laser powers used for different experiments or just for red versus green. Please provide enough information on how data were normalized for someone else to be able to process data in exactly the same way.

We have changed the figure legend for S8. We also included a more detailed description of the normalization procedure in the main methods and have added the macro we ran in Fiji (ImageJ) to the supplemental methods.

Submitted:

- 1. Figure S8: Turnover of the CaMPARI2-red signal.** CA1 neurons from rat hippocampal slices were electroporated at DIV15 with CaMPARI2_F391W-L398V (no epitope tags) and imaged 4 days later on a two-photon microscope. Maximal presynaptic stimulus was elicited by placing a monopolar electrode on the stratum radiatum and giving a 0.2 ms electrical pulse 100 times at 100 Hz. UV light (395 nm, 16 mW mm⁻²) was applied for 2 s with a 1 s delay relative to stimulus (24 neurons, 4 slices). Images were taken before (t = -0.5 h) and immediately after UV + stim. (t=0 h). Slices were put back in the incubator and subsequent images were taken over the next days. A group of neurons were subjected to UV without electrical stimulus as control (21 neurons, 3 slices). Another round of electrical stimuli paired with UV was administered for relabeling of active neurons. Arrows denote administration of stimulus + UV light or UV light alone. The fluorescence intensities, as well as the R/G ratio, were normalized to laser power and plotted. Error bars are standard error of the mean.
- Fiji²⁸ was used for image analysis. Images taken with 2P excitation at 980 nm and 1040 nm were aligned to correct for both chromatic aberration and any slight misalignment of the lasers.²⁹ Regions of interest were drawn onto maximum projections of acquired images after median filtering and background subtraction. For every imaging session, a mixture of fluorescein and sulforhodamine 101 was used to normalize the green and red CaMPARI fluorescence.

Revised:

- 1. Figure S8: Three day turnover of CaMPARI2 and re-conversion. Two-photon image stacks of CA1 neurons from rat hippocampal slice cultures starting 4 days after electroporation at DIV15 with DNA encoding CaMPARI2_F391W-L398V (no epitope tags) at the time points indicated (hours, t = 0 immediately after photoconversion). Arrows indicate when CaMPARI2**

photoconversion was induced by combining strong electrical stimulation of synaptic inputs in the stratum radiatum (100 times at 100 Hz) with UV light (2 seconds, 395 nm, 16 mW mm⁻²) applied with a 1 s delay from the start of stimulation (UV + stim; 24 neurons, 4 slices). Controls received the same UV without electrical stimulation (UV only; 21 neurons, 3 slices). Values are mean ± standard error of the mean.

2. A macro written in Fiji³³ was used for image analysis (see Supplemental Methods). Image stacks taken with 2P excitation at 980 nm and 1040 nm were xyz-aligned to correct for chromatic aberration.³⁴ After median filtering and rolling ball background subtraction, fluorescence values were obtained from regions of interest (ROI) drawn onto maximum intensity projections. As the brightness of CaMPARI versions varies, higher laser power is required to image neurons expressing CaMPARI1 than CaMPARI2_F391W-L398V (no epitope tags) (980nm: 4-10 mW vs 3-5 mW; 1040 nm: 10-15 mW vs 4-8 mW, respectively when measured at the back-focal plane of the objective). To allow comparison between imaging sessions, 1 ml of an aqueous solution containing 2 µg ml⁻¹ fluorescein (Alcon) and 0.2 µg ml⁻¹ sulforhodamine 101 (Tocris) was placed in the imaging chamber after each session and fluorescence intensity was measured with identical acquisition settings used to collect the CaMPARI images in that session. The green and red CaMPARI fluorescence values were normalized by dividing by the values from the calibration solution.

4) Figure 4 was weak and unconvincing. The data presented in S15B are much stronger and more quantitative. Please consider moving S15B to main manuscript.

The purpose of Figure 4 was to demonstrate the unfixed CaMPARI2 signal, the reduced signal after fixation and how the images can be recovered using our anti-CaMPARI2-antibody. Doing this proved to be hard (it requires fixing mounted slices), and is actually an experiment that anti-CaMPARI2-red-antibody users would not do, since we show its specificity.

The purpose of figures S15 and S18, however, was to demonstrate that the anti-FLAG to anti-CaMPARI2-red antibody signals correlate with the endogenous red-to-green ratios. These figures miss the non-fixed images and were thus moved to the supplemental.

Given the comment of the reviewer, we have come to the conclusion that they might indeed be more valuable than we previously thought, so we re-made figures S15 and S18 and turned them into a main Figure 5.

5) Figure 1, bottom, middle graph is not described in the caption.

This panel is referred to in the caption as Panel C, right (as opposed to panel C, left). Since this was not immediately clear to the reviewer, this might be equally unclear to the reader. We thus modified the caption slightly in order to make this more clear.

Submitted:

(C) Photoconversion timecourse showing the red fluorescence of CaMPARI1 (black) and CaMPARI2 (red) in the presence (solid lines) or absence (dashed lines) of calcium in purified protein (left), and with (solid lines) or without (dashed lines) 80 Hz stimulation of primary rat hippocampal neurons (right) as a function of exposure to 405 nm light.

Revised:

(C) Photoconversion timecourse showing the red fluorescence of CaMPARI1 (black) and CaMPARI2 (red) **as a function of exposure to 405 nm light. Left: photoconversion of purified**

CaMPARI protein in the presence (solid lines) or absence (dashed lines) of calcium. Right: photoconversion of primary rat hippocampal neurons with (solid lines) or without (dashed lines) 80 Hz stimulation.

6) Figure 2: please include x-axis for right hand plots. Again, please detail laser powers used and normalization procedure. Y-axis is labeled “fluorescence (AU)” or “fold R/G” but caption implies these have been normalized. Please clarify.

We have added labels to the X-axes and changed the Y axes labels to more accurately indicate that the values were normalized. We also included a more detailed description of the normalization procedure in the main methods and have added the macro we ran in Fiji (ImageJ) to the supplemental methods.

(see comment 3 for changes made to the manuscript)

REVIEWERS' COMMENTS:

Reviewer #1 (Remarks to the Author):

I see that the authors addressed my comments honestly and to the best of their abilities. I believe that authors presented sufficient evidence to convince readers that this presented work is a significant improvement on the previous version of Campari. In addition they also contributed with a novel antibody to improve the use of Campari in several new applications. The tools development is a lengthy process and it is important to support and appreciate the amount of work presented here. Even if the paper do not get into the biological insights of the brain circuits, I agree with the authors that this would be beyond the scope of such a methods paper.

Moreover, I went through the newly added data in response to other reviewers' comments and I appreciate this additional work which certainly justifies the publication of this work in Nature Communications. Hence, I recommend acceptance of this manuscript.

Reviewer #3 (Remarks to the Author):

The authors have addressed all my previous comments (which were minor to begin with). The current iteration is more quantitative and even stronger than before. I am fully supportive of this manuscript and think it will add a valuable tool to the scientific community.